# QONVOLUTION: TOWARDS LEARNING HIGH-FREQUENCY SIGNALS WITH QUERIED CONVOLUTION

## ABSTRACT

Accurately learning high-frequency signals is a challenge in computer vision and graphics, as neural networks often struggle with these signals due to spectral bias or optimization difficulties. While current techniques like Fourier encodings have made great strides in improving performance, there remains scope for improvement when presented with high-frequency information. This paper introduces Queried-Convolutions (Qonvolutions), a simple yet powerful modification using the neighborhood properties of convolution. Qonvolution convolves a low-frequency signal with queries (such as coordinates) to enhance the learning of intricate high-frequency signals. We empirically demonstrate that Qonvolutions enhance performance across a variety of high-frequency learning tasks crucial to both the computer vision and graphics communities, including 1D regression, 2D super-resolution, 2D image regression, and novel view synthesis (NVS). In particular, by combining Gaussian splatting with Qonvolutions for NVS, we showcase state-of-the-art performance on real-world complex scenes, even outperforming powerful radiance field models on image quality. Our code and models will be made publicly available.

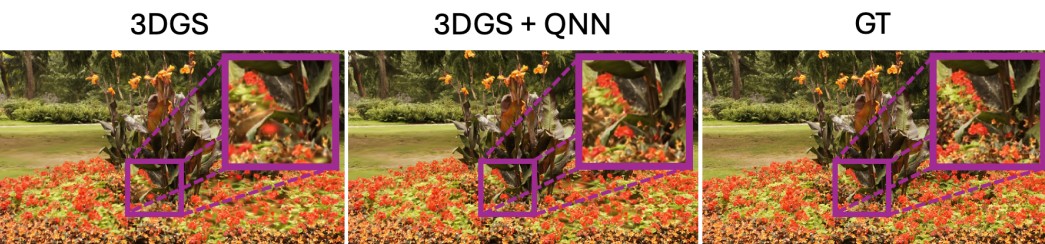

Figure 1: **Learning high-frequency with Qonvolution.** We provide an example on novel view synthesis of 3D Gaussian Splatting (Kerbl et al., 2023) and adding QNN. Adding QNN faithfully reconstructs high-frequency details in various regions and results in higher quality synthesis. We highlight the differences in inset figures.

## 1 INTRODUCTION

Neural networks are now fundamental to computer vision and graphics, for deciphering a wide range of signals, from intricate 1D data like time series (Kazemi et al., 2019) and natural language (Vaswani et al., 2017) to rich 2D images (Tancik et al., 2020), and immersive 3D scenes (Barron et al., 2023). However, these networks often struggle to capture high-frequency details, a challenge often attributed to spectral bias (Rahaman et al., 2019) or optimization difficulties (Tab. 9) due to complicated landscapes (Li et al., 2018).

The challenge of capturing high-frequency details in neural networks has spurred a rich and diverse body of research. A key strategy involves modifying positional encodings, using Fourier encodings (Tancik et al., 2020). Another approach focuses on altering activation functions, such as those in SIREN (Sitzmann et al., 2020). Other methods aim to predict Fourier series coefficients (Lee et al., 2021), use high-frequency weighted losses (Zhang et al., 2024) or tune weight initialization (Saratchandran et al., 2024). While all these innovations improve performance, the spectrum of frequencies effectively learned remains limited, highlighting a compelling need for further advancements in high-frequency signal representation.

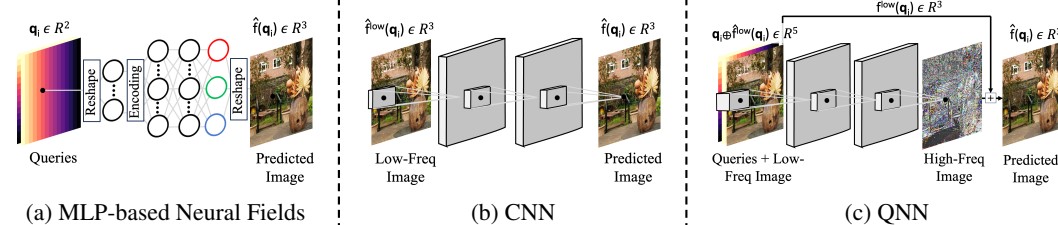

|  | | |
|---|---|---|
| (a) MLP-based Neural Fields | (b) CNN | (c) QNN |

Figure 2: **Overview** of MLPs, CNNs and QNNs. MLPs take (encoded) queries $\gamma(\mathbf{q}_i)$ and uses linear layers. CNNs take the low-frequency signal $\hat{f}^{low}$ and uses convolutions. QNN concatenates the low-frequency signal $\hat{f}^{low}$ to the (encoded) queries $\gamma(\mathbf{q}_i)$ and uses convolutions. [Key: Freq = Frequency, $\oplus$= Concatenation]

There are two classes of tasks that attempt to learn high-frequency signals. One stream, which includes the popular novel view synthesis (NVS) task, uses Multi-Layer Perceptrons (MLPs) (Fig. 2a) to directly fit signals. However, MLPs lack the necessary inductive biases (Cohen & Welling, 2016; LeCun et al., 1998) to capture the local neighborhood dependencies inherent in most 1D and 2D signals. We conjecture that these local relationships are crucial for learning high-frequency information effectively. By processing data points or pixels in isolation, existing MLP networks often neglect these local connections, which limits their capacity to fully represent high-frequency signals. On the other hand, if a low-frequency signal is given, the second class of problems, which includes, *e.g.*, 2D super-resolution (SR), convolves the low-frequency signal with a CNN (Fig. 2b), and thus, uses neighborhood information. However, these approaches (Karras et al., 2018) often do not utilize the information present in the input queries (*e.g.*: spatial coordinates) in predicting the output high-frequency signal. As shown in previous work (Liu et al., 2018), architectures that are aware of locality, such as CNNs, often fail at tasks that require even a simple transformations of coordinates. Thus, the potential for queries to also contribute to the reconstruction of high-frequency details in such tasks remains an area for further investigation.

To address the limitations of existing methods and effectively leverage neighborhood dependencies, valuable low-frequency signals and queries, this paper introduces Queried-Convolution, or Qonvolution. As a building block, it replaces the traditional linear layer of MLPs with a convolutional layer and processes queries alongside a low-frequency signal. This approach, which convolves a low-frequency signal with queries, marks a departure from current methodologies in neural fields (see Fig. 2). We rigorously evaluate[1] Qonvolution Neural Networks (QNNs) (Fig. 2c) across a diverse range of high-frequency learning tasks, including 1D regression (Fig. 3), 2D SR, 2D residual image regression, and NVS (Fig. 1). Our extensive experiments consistently demonstrate that QNNs significantly outperform all established baselines. Notably, a QNN with a 3D Gaussian Splatting (3DGS) based baseline surpasses even the Zip-NeRF (Barron et al., 2023) model in the challenging NVS task, while training faster than Zip-NeRF.

In summary, this paper makes the following contributions:

- We propose QNNs, an architecture that convolves both low-frequency signals and queries. This approach improves the learning of high-frequency details.
- We also theoretically investigate the predictive power of QNNs compared to CNNs (Theorem 1).
- Our extensive experiments show that QNNs enhance performance across diverse tasks, including 1D regression (Sec. 5.1), 2D SR (Sec. 5.2), 2D image regression (Sec. 5.3), and NVS (Sec. 5.4).

## 2 LITERATURE REVIEW

**Overcoming Spectral Bias.** The inherent "spectral bias" of MLPs (Rahaman et al., 2019), where they favor learning low-frequency components over fine-grained, high-frequency details, poses a significant hurdle in representing complex signals. A closely related phenomenon is the low-rank bias (Huh et al., 2022). To combat these limitations, the literature presents a variety of strategies. The first prominent direction involves modifying positional encodings. Techniques like sinusoidal encodings (Vaswani et al., 2017; Mildenhall et al., 2020), Fourier encodings (Rahimi & Recht, 2007; Tancik et al., 2020) and hash-grids (Müller et al., 2022) provide MLPs with more discriminative spatial information, which helps them learn higher frequencies. Similarly, Yu et al. (2025)

---

[1]Except for SR, this paper mostly covers coordinate-based networks (neural fields) for regression, where the focus is fitting to a single signal than in generalizable or amortized models.

feed wavelet-decomposed signals into neural networks, leveraging the multi-resolution properties of wavelets. Another direction centers on redesigning activation functions to boost the MLP capacity for high-frequency representation. Examples include the use of the error function (erf) (Yang & Salman, 2019), the periodic activations in SIREN (Sitzmann et al., 2020), sinc functions (Saratchandran et al., 2024), FINER (Liu et al., 2024) or QIREN functions (Zhao et al., 2024). Some methods also work directly in the frequency domain, either by predicting Fourier series coefficients (Lee et al., 2021) or phase-shifted signals (Cai et al., 2020). High-frequency weighted losses (Zhang et al., 2024) is another effective strategy, that forces the network to reconstruct fine-grained details during training. Additional efforts to overcome spectral bias include carefully tuning weight initialization (Saratchandran et al., 2024; Teney et al., 2024) for more balanced learning across all frequencies. Some methods employ neural network ensembles (Ainsworth & Dong, 2022; Wang & Lai, 2024) to collectively capture high-frequency information. Compared to these prior methods, which primarily use MLPs over queries, this paper convolves both queries and the low-frequency signal. This leverages neighborhood dependencies and the low-frequency representation for superior high-frequency signal learning.

**Image Super-Resolution (SR).** The SR literature is seeing a paradigm shift from Generative Adversarial Networks (GANs) (Wang et al., 2021) to diffusion models (Saharia et al., 2022; Yue et al., 2023), primarily building on CNNs (Wang et al., 2021), with some exceptions (Lee & Jin, 2022). To achieve arbitrary resolution, implicit representations are a key strategy (Chen et al., 2021; Lee & Jin, 2022). In this context, QNN stands apart by integrating queries and encoded images through convolutions, contrasting with MLP-based processing of queries and latents. While Lee & Jin (2022) also use CNNs and queries, their core innovation centers on predicting Fourier space coefficients in the output, whereas QNN focuses on the convolutional mechanism for query integration.

**Novel-View Synthesis (NVS).** Neural Radiance Fields (NeRFs) represent a 3D scene as a continuous function for NVS; however, their reliance on ray-marching results in slow training and rendering (Mildenhall et al., 2020; Barron et al., 2022). Subsequent research addresses these limitations by introducing better encodings (Müller et al., 2022; Tancik et al., 2020) and multi-scale representations (Barron et al., 2023). In contrast, 3DGS renders a scene with rasterized 3D gaussians, offering faster training and real-time rendering. Subsequent works improve reconstruction quality through better initialization (Wang et al., 2025), exploration (Kheradmand et al., 2024), primitives (Arunan et al., 2025; Li et al., 2025a), multi-scale representations (Yu et al., 2024), better projections (Huang et al., 2024) and frequency-weighting (Zhang et al., 2024). A few works integrate ray-marching with 3DGS (Loccoz et al., 2024), though this typically sacrifices the speed gains. Other studies attempt to generalize NeRFs or 3DGS to multiple scenes (Charatan et al., 2024; Schnepf et al., 2025) or improve efficiency (Aumentado-Armstrong et al., 2023) with CNNs. Conversely, this work applies QNNs to rasterized 3DGS images to achieve superior high-fidelity view synthesis. Finally, video diffusion models like CAT3D (Gao et al., 2024) and LVSM (Jin et al., 2025) synthesize novel views by concatenating image latents and 3D queries. In contrast, QNN focuses on established NVS benchmarks, rather than sparse-view or generative modelling. Although PRoPE (Li et al., 2025b) and MVGD (Guizilini et al., 2025) concatenate images and queries, they neither use convolutional architectures, nor show that concatenation helps learning high-frequency signal. PaDIS (Hu et al., 2024) concatenates image coordinates to image as input to a diffusion model.

**Theoretical Results.** Many theoretical works demonstrate that CNNs exhibit superior sample complexity compared to MLPs, for both binary classification (Bietti, 2021; Li et al., 2021) and regression problems (Du et al., 2018; Misiakiewicz & Mei, 2022; Wang & Wu, 2023), provided certain conditions are met. Many common computer vision tasks, including NVS, meet these conditions (Ulyanov et al., 2018). Some attribute the enduring utility of the CNN to its inductive biases, specifically weight sharing (Bruna & Mallat, 2013; Lenc & Vedaldi, 2015; Mei et al., 2021) and neighborhood (Pogodin et al., 2021; Malach & Shalev-Shwartz, 2021; Lahoti et al., 2024). For example, Malach & Shalev-Shwartz (2021) showed that CNNs learn more accurate mappings than MLPs when the classification label depends on the neighborhood information. To the best of our knowledge, no theoretical results directly compare QNNs with CNNs and MLPs in regression settings.

## 3 PRELIMINARIES

We begin by establishing the notations and basics of neural fields in the following paragraphs.

**Neural Fields.** A neural field, denoted as $f_\theta : \mathcal{Q} \to \mathcal{O}$, with parameters $\theta$, represents a signal by mapping a bounded set of queries $\mathcal{Q} \subset \mathbb{R}^d$ to outputs $\mathcal{O} \in \mathbb{R}^m$. Specifically, for an input query vector $\mathbf{q}_i \in \mathcal{Q}$, the neural field produces an output $f_\theta(\mathbf{q}_i) \in \mathcal{O}$. This framework is highly versatile, accommodating diverse signal types and queries, such as 1D audio, 2D images, or 3D geometry. Illustrative examples of tasks, inputs and their corresponding outputs include:

- 1D Regression: 1D coordinate queries $\mathbf{q}_i \in \mathbb{R}^1$, outputting 1D scalar values (Tancik et al., 2020);
- 2D Image Regression: 2D coordinate queries $\mathbf{q}_i \in \mathbb{R}^2$, representing pixel locations, outputting 3D RGB pixel colors (Stanley, 2007);
- NVS with NeRF: 5D queries[2] $\mathbf{q}_i \in \mathbb{R}^5$, combining 3D camera coordinates with 2D view directions, outputting 4D color and density information (Mildenhall et al., 2020).

Most neural fields commonly employ MLPs to implement the function $f_\theta$. To construct high-frequency details, they first pass a query $\mathbf{q}_i$ through an encoding $\gamma$ such as Fourier encodings (Tancik et al., 2020) and hash-grids (Müller et al., 2022). So, we write the MLP-based neural field as

$$f_\theta^{MLP}(\mathbf{q}_i) = MLP(\gamma(\mathbf{q}_i)). \tag{1}$$

Vanilla encoding refers to $\gamma(\mathbf{q}_i)=\mathbf{q}_i$. The neural field processes $N$ input samples $\mathbf{q}_i$, each with $d=C_{in}$ input dimensions (channels), $C_h$ hidden dimensions (channels) and generates $N$ output samples with $m=C_{out}$ output dimensions (channels). Consequently, the neural field transforms a tensor of shape $N \times C_{in}$ into an output tensor of shape $N \times C_{out}$. One then reshapes the output samples to match the target signal dimensions.

**Loss.** One optimizes the parameters $\theta$ of these networks by minimizing a loss, such as the squared loss between the network's output and the true signal, over the observed data.

**Example.** As an example, consider a 2D image regression task where 2D coordinates serve as queries. In this scenario, the input channels $C_{in}$ are 2, representing the $(u, v)$ pixel coordinates, and the output channels $C_{out}$ are 3, corresponding to the RGB pixel colors. For each image, the number of data samples, $i$, ranges from 1 to $N$, where $N=H \times W$ (height times width of the image).

## 4 QUERIED-CONVOLUTION NEURAL NETWORKS (QNN)

Having established the necessary notations and preliminaries in the preceding section, we now introduce our proposed Qonvolution and QNN in this section.

### 4.1 QNN

Let $\hat{f}^{low}$ and $f^{low}$ respectively be the learned and the GT low-frequency approximation of the true signal $f$. Note that this low-frequency signal is often available in certain tasks: for example, the NVS task can use the splatted 3DGS image as the learned low-frequency signal, while the SR task itself provides a GT low-pass image as the input.

We hypothesize that there is benefit to providing the low-frequency approximation of neighboring coordinates. So, we set out to design a neural field which exploits the information present in the neighborhood of the given query $\mathbf{q}$ in terms of the query values and the corresponding low-frequency approximations. Then, the Qonvolution neural network (QNN) operates by concatenating the neighborhood of encoded input queries $\gamma(\mathbf{q}_{\mathcal{N}(i)})$ with their low-frequency signal $\hat{f}^{low}_{\mathcal{N}(i)}$, and subsequently convolving them to produce the output $f_\theta(\mathbf{q}_i) \in \mathcal{O}$, with $\mathcal{N}(i)$ denoting the neighbors of index $i$ including itself. We formally write the QNN as

$$f_\theta^{QNN}(\mathbf{q}_i) = CNN(\gamma(\mathbf{q}_{\mathcal{N}(i)}) \oplus \hat{f}^{low}_{\mathcal{N}(i)}), \tag{2}$$

with $\oplus$ denoting the concatenation of tensors along the channel dimensions.

A distinct advantage of the QNN architecture is that it needs to be evaluated once for the entire signal, rather than once per coordinate. However, a downside is QNNs can not be used in tasks without neighborhoods such as casting a single ray in NeRF. Furthermore, the QNN architecture offers considerable flexibility in the choice and integration of queries; various queries can be appended depending on the specific task at hand. We refer to our ablation studies (Sec. 5.5), where we experiment with different queries.

---

[2]In practice, NeRF uses 6D queries with 3D view directions, but the norm of 3D view direction is 1.

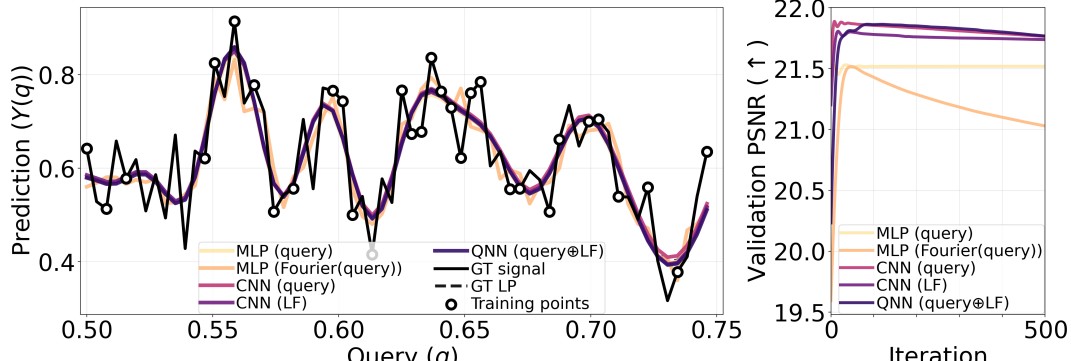

Figure 3: **1D Regression.** QNN **outperforms** MLP-based architectures including Fourier encodings in regressing high-frequency signals. This simple experiment compares networks which take the 1D queries and low-frequency (LF) signal as input to predict the high-frequency 1D signal. The standard MLP-based networks including Fourier encodings take 1D coordinates as queries. QNN changes the linear layer to a 1D convolutional layer and also takes the low-frequency signal in addition to the 1D queries. [Key: ⊕= Concatenation].

**Example.** To illustrate, consider the same 2D image regression task previously discussed in Sec. 3, characterized by $C_{in}$=2 input channels and $C_{out}$=3 output channels. The number of data samples for each image is $N$=$H{\times}W$. In this context, the low-frequency image is a tensor of shape $C_{low}{\times}H{\times}W$. The QNN appends both this low-frequency tensor and the 2D coordinate queries. Consequently, the QNN receives an input tensor of shape $(C_{low} + C_{in}){\times}H{\times}W$, or $5{\times}H{\times}W$, and outputs a $C_{out}{\times}H{\times}W$, or $3{\times}H{\times}W$ tensor without reshaping.

**Remarks.** The QNN generalizes several existing architectures in the literature:
- Replacing the CNN with an MLP and excluding the low-frequency signal $\hat{f}^{low}$, reduces the QNN to conventional neural fields.
- By omitting the input queries $\mathbf{q}_i$, the QNN effectively simplifies to a standard CNN architecture.
- When utilizing normalized 2D coordinates as queries with vanilla encodings and employing 2D convolutions, QNN becomes the coordinate CNN (Liu et al., 2018).

### 4.2 Comparing QNN, CNN and MLP on Regression Tasks

In this section, we provide theoretical justification for our approach with real-valued target functions. We show that, in the setting which is not limited by data nor computation, adding contextual information, specifically neighborhood information and then the neighboring queries, does not negatively impact the achievable risk when predicting from a low-frequency signal. Furthermore, we show that, in this setting, adding queries guarantees that we can perfectly approximate the target function, achieving the best possible error.

**Theorem 1.** *Let $\mathcal{X}$ be an input space, which is a discrete integer lattice or in $\mathbb{R}^d$, and let $P_{\mathcal{X}}$ be a probability distribution on $\mathcal{X}$. Let $f : \mathcal{X} \to [0,1]$ be a deterministic and measurable target function. A feature map is a deterministic, measurable function $\phi : \mathcal{X} \to \mathcal{Z}$ that maps inputs to a feature space $\mathcal{Z}$. The corresponding hypothesis class, $\mathcal{H}(\phi)$, is the set of all deterministic, measurable functions mapping from $\mathcal{Z}$ to the output space $[0,1]$. The optimal expected error for a given feature map $\phi$ is the minimum mean squared error achievable by any hypothesis in its class $\mathcal{R}^*(\phi, f) = \min_{h \in \mathcal{H}(\phi)} \mathbb{E}\left[(f(x) - h(\phi(x)))^2\right]$. Let $f^{low}{:}\mathcal{X}{\to}[0,1]$ be a deterministic measurable function that approximates $f$. Consider the sequence of increasingly informative feature maps:*
- *Feature Map 1 (Approximation) or MLP (LF): $\phi_1(x) = (f^{low}(x))$*
- *Feature Map 2 (Neighborhood) or CNN (LF): $\phi_2(x) = (\phi_1(x - \Delta), \phi_1(x), \phi_1(x + \Delta))$*
- *Feature Map 3 (Neighborhood and Queries) or QNN: $\phi_3(x) = (\phi_2(x), x, x - \Delta, x + \Delta)$,*

*for a fixed $\Delta > 0$. Then, the optimal expected errors for these feature maps are monotonically non-increasing and bounded by zero:*

$$\mathcal{R}^*(\phi_1, f) \geq \mathcal{R}^*(\phi_2, f) \geq \mathcal{R}^*(\phi_3, f) = 0 \tag{3}$$

The proof (Sec. B.1) utilizes optimal squared error predictors (Bishop & Nasrabadi, 2006) and the non-increasing error property with added random variables (Xu & Raginsky, 2022).

Table 1: **2D Image SR Results. Adding QNN outperforms** baselines across both SR sub-tasks. [Key: **Best**, ^⋄= Reported, ^*= Retrained].

| Method | QNN | Face SR | | Natural SR | |
|---|---|---|---|---|---|
| | | PSNR (↑) | SSIM (↑) | PSNR (↑) | SSIM (↑) |
| PULSE (Menon et al., 2020) | − | 16.88 | 0.44 | − | − |
| FSRGAN (Chen et al., 2018) | − | 23.01 | 0.62 | − | − |
| SR3^⋄ (Saharia et al., 2022) | ✗ | 23.04 | 0.65 | − | − |
| SR3^* (Saharia et al., 2022) | ✗ | 23.51 | 0.68 | − | − |
| SR3 (Saharia et al., 2022) | ✓ | **23.63** (+0.12) | **0.69** (+0.01) | − | − |
| Real-ESRGAN^⋄ (Wang et al., 2021) | ✗ | − | − | 24.84 | 0.71 |
| Real-ESRGAN^* (Wang et al., 2021) | ✗ | − | − | 24.41 | 0.70 |
| Real-ESRGAN (Wang et al., 2021) | ✓ | − | − | **24.63** (+0.22) | 0.70 (+0.00) |

## 5 EXPERIMENTS

We evaluate on four high-frequency tasks crucial to both the computer vision and graphics communities, including 1D regression, 2D SR, 2D image regression, and NVS. The 1D regression, 2D regression and NVS experiments compares QNNs against coordinate-based networks (neural fields) and CNNs, while 2D SR experiments compares QNNs against CNNs.

### 5.1 1D REGRESSION

We begin by comparing the performance of MLPs, CNNs and our proposed QNNs on a 1D regression task involving a high-frequency target signal. The networks take 1D coordinates (and low-frequency signal) as input to produce 1D output.

**Dataset.** Our experiments use the synthetic 1D $1/f^\alpha$ signal, defined on the interval $[0, 1)$, with $\alpha = 0.5$ (Tancik et al., 2020). We then apply a low-pass filter with a normalized cutoff of $0.125$ to separate this signal into its low- and high-frequency components, and our task is regressing only the high-frequency component. More details are in Sec. C.1.

**Data Splits and Evaluation Metrics.** For training and testing, we randomly sample data points, using a signal length $N = 32$ (Tancik et al., 2020). We calculate the loss on the training samples to avoid data leakage. We use the Peak Signal-to-Noise Ratio (PSNR) metric for evaluation.

**Implementation and Baselines.** We use MLP with vanilla and 256-dimensional Fourier encodings (Tancik et al., 2020) as our baselines. The MLP architecture consists of 4 linear layers with 256 dimensions and ReLUs (Tancik et al., 2020) to output 1D signal. The MLP takes 1D queries as input, and so we replace the linear layer in MLP with 1D convolution to construct CNN and QNN. Note that the QNN takes both the 1D coordinates and low-frequency component as inputs.

**Results.** Fig. 3 confirms that all CNN models outperform MLP architectures in the PSNR metric, including those with Fourier encodings, at regressing the high-frequency signal.

### 5.2 2D IMAGE SUPER-RESOLUTION (SR)

We further evaluate the efficacy of our proposed QNN on the image SR task, a critical application that reconstructs high-resolution (HR) images by adding high-frequency details to the existing low-resolution (LR) inputs. Our experimental setup encompasses two distinct image SR scenarios:
- Face SR: This sub-task involves $8\times$ upscaling of face images from $16\times16$ to $128\times128$ resolution. It is trained on Flickr-Faces-HQ (FFHQ) and evaluated on CelebA-HQ (Saharia et al., 2022).
- Natural SR: This sub-task involves $4\times$ upscaling of natural images from $64\times64$ to $256\times256$ resolution on DIV2K (Wang et al., 2021).

**Datasets and Splits.** For face SR, we leverage the FFHQ dataset (Karras et al., 2019), utilizing all its 70,000 images for training, and test on 30,000 CelebA-HQ dataset (Karras et al., 2018) as in Saharia et al. (2022). For natural SR, our training follows Real-ESRGAN (Wang et al., 2021), comprising 13,774 HR images, with 800 train images from DIV2K (Agustsson & Timofte, 2017), 2,650 images from Flikr2K (Timofte et al., 2017) and 10,324 images from the OST dataset (Wang et al., 2018). We evaluate on the paired 100 DIV2K validation images.

**Evaluation Metrics.** We use the widely accepted PSNR and Structural Similarity Index Metric (SSIM) (Wang et al., 2004) metrics to quantitatively asses the SR images (Saharia et al., 2022).

Table 2: **2D Residual Image Regression Results. Adding QNN outperforms** baselines on this regression task. [Key: **Best**].

| Encoding ($\gamma$) | QNN | Val | | Train | |
|---|---|---|---|---|---|
| | | PSNR (↑) | SSIM (↑) | PSNR (↑) | SSIM (↑) |
| Low-frequency images | − | 29.61 | 0.89 | 29.65 | 0.89 |
| Hash-grid (Müller et al., 2022) | ✗ | 29.35 | 0.88 | 30.34 | 0.88 |
| Vanilla | ✗ | 29.69 | 0.92 | 29.69 | 0.92 |
| | ✓ | **32.48** (+2.79) | **0.96** (+0.04) | **32.51** (+2.82) | **0.96** (+0.04) |
| Fourier (Tancik et al., 2020) | ✗ | 29.79 | 0.92 | 29.80 | 0.92 |
| | ✓ | **32.87** (+3.09) | **0.94** (+0.02) | **32.93** (+3.13) | **0.94** (+0.02) |

**Implementation and Baselines.** To thoroughly assess the impact of our proposed QNN, we integrate it into two prominent architectural paradigms: diffusion models and GANs. Specifically, for face SR, we use an unofficial implementation[3] of SR3 (Saharia et al., 2022), a DDPM-based SR method. For natural SR, we leverage the official Real-ESRGAN (Wang et al., 2021) codebase[4], which is a GAN-based SR method. It is crucial to note that both the original SR3 and Real-ESRGAN models use CNN architectures and take the LR input. We integrate QNN into both SR3 and Real-ESRGAN, which serve as our baselines and train with exactly same iterations, loss functions and other hyperparameters. We realise QNN by augmenting these existing CNN models with 2D coordinate queries. Note that QNN has slightly more number of parameters (0.0068% more for Real-ESRGAN generator) than CNN because input to the QNN is a 5-channel image (3 RGB + 2 coordinates) compared to 3-channel RGB image as input to the CNN.

**Results.** Tab. 1 presents the results of our image SR experiments across both sub-tasks. The findings confirm that the integration of QNNs consistently enhance the ability of these models to learn and reconstruct high-frequency details for both SR sub-tasks. Interestingly, the observed PSNR gain for face SR (0.12 PSNR) is comparatively smaller than that for natural SR (0.22 PSNR). This outcome aligns with our expectations, as face images are generally considered to be inherently lower-rank in terms of their frequency content when compared to the richer and more varied textural information present in natural images. We also show some qualitative results in Fig. 5 in the supplementary.

## 5.3 2D RESIDUAL IMAGE REGRESSION

We next evaluate it on the 2D residual image regression task, where the networks take the 2D coordinates (and low-frequency image) as input to output three-dimensional RGB values.

**Datasets.** We use the Mip-NeRF 360 dataset (Barron et al., 2022) for this experiment, and the rendered output from the baseline 3DGS (Kerbl et al., 2023) as the low-frequency image input to the QNN. The QNN learns to predict the residual error left by 3DGS with respect to the GT image. Specifically, the overall prediction is calculated as the 3DGS-splatted image (low-frequency baseline) plus the QNN's predicted residual image (high-frequency correction). The loss is then computed between the summed prediction and the GT Mip-NeRF 360 image.

**Data Splits and Evaluation Metrics.** We randomly sample training and validation pixels over images. We use the PSNR and SSIM (Wang et al., 2004) metrics for evaluation as (Tancik et al., 2020) over both validation and training pixels. We run over all scenes of the Mip-NeRF 360 dataset and average scores across scenes to report a single value for each network.

**Implementation and Baselines.** The MLP architecture has 4 linear layers, 256 dimensions and ReLUs to output three-dimensional RGB signal. The MLP takes 2D queries as input, and so, we replace the linear layer in MLP with 2D convolution and append low-frequency image to construct QNN. We use QNN with both Vanilla and Fourier encodings, and use the MLP with vanilla encodings, fourier encodings (Tancik et al., 2020) and hash-grids (Müller et al., 2022) as our baselines.

**Results.** Tab. 2 shows the results on 2D image regression task on both val and train splits. Tab. 2 confirms that adding QNN benefits learning high-frequency details for 2D residual image regression task on both val and train splits.

---

[3]https://github.com/Janspiry/Image-Super-Resolution-via-Iterative-Refinement SR3 authors do not release their official code. We do not report SR3's performance on natural SR since this unofficial code does not reproduce natural SR results.

[4]https://github.com/xinntao/Real-ESRGAN

Table 3: **NVS Results. Adding QNN outperforms** the baselines across all datasets. Numbers are from respective papers. Stable-GS, 3D-HGS and MCMC do not report results on all nine scenes of Mip-NeRF 360 dataset. [Key: Best , Second-best , △= Reported, *= Retrained, †= Reported in 3DGS (Kerbl et al., 2023)].

| Render | Method | Venue | Mip-NeRF 360 PSNR↑/SSIM↑/LPIPS↓ | Tank & Temples PSNR↑/SSIM↑/LPIPS↓ | Deep Blending PSNR↑/SSIM↑/LPIPS↓ | OMMO PSNR↑/SSIM↑/LPIPS↓ | Shelly PSNR↑/SSIM↑/LPIPS↓ | Syn-NeRF PSNR↑/SSIM↑/LPIPS↓ |
|---|---|---|---|---|---|---|---|---|
| Ray | Plenoxels† (Keil et al., 2022) | CVPR22 | 23.08 / 0.63 / 0.44 | 21.08 / 0.72 / 0.38 | 23.06 / 0.80 / 0.51 | – | – | 31.76 / – / – |
| | INGP-Big† (Müller et al., 2022) | SIGG22 | 25.59 / 0.75 / 0.30 | 21.92 / 0.75 / 0.31 | 24.96 / 0.82 / 0.39 | – | – | 33.18 / – / – |
| | Adaptive (Wang et al., 2023) | SIGG23 | – | – | – | – | 36.02 / 0.95 / 0.08 | 32.51 / 0.96 / 0.05 |
| | QFields (Sharma et al., 2024) | ECCV24 | – | – | – | – | 37.29 / 0.95 / 0.07 | 31.00 / 0.95 / 0.07 |
| | 3DGRT (Loccoz et al., 2024) | SIGG25 | 27.20 / 0.82 / 0.25 | 23.20 / 0.83 / 0.22 | 29.23 / 0.90 / 0.32 | – | – | – |
| | 3DGUT (Wu et al., 2025) | CVPR25 | 27.26 / 0.81 / 0.22 | 22.90 / 0.84 / 0.17 | – | – | – | – |
| | VKRayGS (Bulò et al., 2025) | CVPR25 | 27.27 / 0.82 / 0.22 | – | – | – | – | – |
| | EVER (Mai et al., 2025) | ICCV25 | 27.51 / 0.83 / 0.23 | – | – | – | – | – |
| | Mip-NeRF† (Barron et al., 2021) | CVPR22 | 27.69 / 0.79 / 0.24 | 22.22 / 0.76 / 0.26 | 29.40 / 0.90 / 0.25 | – | – | – |
| | 3DGEER (Huang et al., 2025) | ArXiv25 | 27.76 / 0.82 / 0.21 | – | – | – | – | – |
| | Zip-NeRF (Barron et al., 2023) | ICCV23 | 28.54 / 0.83 / 0.19 | – | – | – | – | – |
| Raster | GES (Hamdi et al., 2024) | CVPR24 | 26.91 / 0.79 / 0.25 | 23.35 / 0.84 / 0.20 | 29.68 / 0.90 / 0.25 | – | – | – |
| | Stable-GS (Wang et al., 2025) | ArXiv25 | – | 24.04 / 0.86 / 0.16 | 29.66 / 0.91 / 0.24 | – | – | – |
| | Convex (Held et al., 2025) | CVPR25 | 27.29 / 0.80 / 0.21 | 23.95 / 0.85 / 0.16 | 29.81 / 0.90 / 0.24 | – | – | – |
| | Vol3DGS (Talegaonkar et al., 2025) | CVPR25 | 27.30 / 0.81 / 0.21 | 23.74 / 0.85 / 0.17 | 29.72 / 0.91 / 0.25 | – | – | – |
| | Textured-GS (Chao et al., 2025) | CVPR25 | 27.35 / 0.83 / 0.19 | 24.26 / 0.85 / 0.17 | 28.33 / 0.89 / 0.36 | – | – | 33.24 / 0.97 / 0.04 |
| | DARB (Arunan et al., 2025) | ArXiv25 | 27.45 / 0.81 / 0.21 | 23.64 / 0.85 / 0.17 | 29.63 / 0.90 / 0.24 | – | – | – |
| | 3D-HGS (Li et al., 2025a) | CVPR25 | – | 25.08 / 0.84 / 0.14 | 29.80 / 0.90 / 0.25 | – | – | – |
| | Project (Huang et al., 2024) | ECCV24 | 27.48 / 0.82 / 0.21 | 23.43 / – / – | 29.51 / – / – | – | – | – |
| | VDGS (Malarz et al., 2025) | CVIU25 | 27.64 / 0.81 / 0.22 | 24.02 / 0.85 / 0.18 | 29.54 / 0.91 / 0.24 | – | – | 35.97 / 0.99 / 0.01 |
| | Revise (Bulò et al., 2024) | ECCV24 | 27.70 / 0.82 / 0.22 | 24.10 / 0.86 / 0.18 | 29.64 / 0.91 / 0.30 | – | – | – |
| | HyRF (Wang & Xu, 2025) | NIPS25 | 27.78 / 0.82 / 0.21 | 24.02 / 0.84 / 0.18 | 30.37 / 0.91 / 0.24 | – | – | – |
| | Mip-Splat (Yu et al., 2024) | CVPR24 | 27.79 / 0.83 / 0.20 | – | – | – | – | – |
| | FreGS (Zhang et al., 2024) | CVPR24 | 27.85 / 0.83 / 0.21 | 23.96 / 0.85 / 0.18 | 29.93 / 0.90 / 0.24 | – | – | – |
| | 3DGS△ (Kerbl et al., 2023) | SIGG23 | 27.20 / 0.82 / 0.21 | 23.14 / 0.84 / 0.18 | 29.41 / 0.90 / 0.24 | – | – | 33.31 / – / – |
| | 3DGS* (Kerbl et al., 2023) | SIGG23 | 27.67 / 0.82 / 0.20 | 23.52 / 0.84 / 0.18 | 29.50 / 0.90 / 0.24 | 29.03 / 0.90 / 0.16 | 37.56 / 0.96 / 0.08 | 34.83 / 0.98 / 0.03 |
| | 3DGS + QNN | – | 27.96 / 0.83 / 0.20 | 24.11 / 0.85 / 0.17 | 29.67 / 0.91 / 0.25 | 29.37 / 0.91 / 0.15 | 37.99 / 0.96 / 0.07 | 35.36 / 0.98 / 0.03 |
| | MCMC△ (Kheradmand et al., 2024) | NIPS24 | – | 24.29 / 0.86 / 0.19 | 29.67 / 0.89 / 0.32 | 29.52 / 0.91 / 0.20 | – | 33.80 / 0.97 / 0.04 |
| | MCMC* (Kheradmand et al., 2024) | NIPS24 | 28.26 / 0.84 / 0.17 | 24.53 / 0.87 / 0.13 | 29.32 / 0.91 / 0.24 | 30.10 / 0.92 / 0.12 | 35.14 / 0.94 / 0.09 | 36.14 / 0.98 / 0.02 |
| | MCMC + QNN | – | 28.58 / 0.84 / 0.16 | 24.87 / 0.88 / 0.12 | 29.76 / 0.91 / 0.23 | 30.34 / 0.92 / 0.12 | 35.57 / 0.95 / 0.09 | 36.58 / 0.98 / 0.02 |

## 5.4 NOVEL VIEW SYNTHESIS (NVS)

Motivated by our residual image regression results, which demonstrated the potential to close the gap in image quality between NeRF-based and 3DGS-based methods, we next considered the challenging NVS task. NVS involves rendering a scene from a new viewpoint, based on a finite set of input images and associated poses. This involves training QNN with the baseline end-to-end.

**Datasets.** Our NVS experiments use six diverse datasets with a total of 35 scenes: Mip-NeRF 360 (Barron et al., 2022), Tank & Temples (Knapitsch et al., 2017), Deep Blending (Hedman et al., 2018), OMMO (Lu et al., 2023), Shelly (Wang et al., 2023) and Syn-NeRF (Mildenhall et al., 2020). This selection includes a mix of synthetic, real, bounded indoor, and unbounded outdoor scenes to ensure a comprehensive evaluation. Specifically, we use all nine scenes from Mip-NeRF 360, the same two scenes each from Tank & Temples and Deep Blending (as in Kheradmand et al. (2024)), and all eight scenes from Syn-NeRF, consistent with the 3DGS (Kerbl et al., 2023). We also test on all eight scenes from the OMMO dataset, which provides scenes with distant objects (Kheradmand et al., 2024) and all six scenes from the Shelly dataset, known for its fuzzy surfaces and complex geometries (Sharma et al., 2024).

**Data Splits.** We take every 8th image for test and remaining as train for Mip-NeRF 360, Tank & Temples, Deep Blending and OMMO datasets, as in Barron et al. (2022). We use the provided train/test JSONs for the Shelly and Syn-NeRF datasets. More details are in Sec. C.4.

**Evaluation Metrics.** We evaluate each method using three standard metrics: PSNR, SSIM (Wang et al., 2004) and VGG-16 based normalized Learned Perceptual Image Patch Similarity (LPIPS) (Zhang et al., 2018) as in Kerbl et al. (2023). We average the scores across all scenes to report a single value for each dataset.

**Implementation and Baselines.** We integrate QNN into GS-based baselines (Fig. 7) since these methods are fast to train/test compared to the NeRF-based methods. We integrate the QNN into both 3DGS (Kerbl et al., 2023) and MCMC (Kheradmand et al., 2024), which serve as our baselines. Additionally, we also compare against integrating MLP-based neural fields instead of QNN. See Sec. C.4 for details.

**Results.** Tab. 3 shows the NVS results on all six datasets. Tab. 3 confirms that adding a QNN outperforms all the baselines and significantly benefits learning high-frequency signals for the challenging NVS task. Notably, a QNN with the MCMC baseline surpasses even Zip-NeRF (Barron et al., 2023) in this task, a testament to its effectiveness. We show some qualitative results in Fig. 6.

Table 4: **Ablation Studies** of QNN on the NVS task with 3DGS (Kerbl et al., 2023) on the Mip-NeRF 360 dataset. QNN outperforms all MLP in fitting training data. [Key: Best , ⊕= Concatenation, LF = Low-Frequency]

| Change | From → To | Val | | | Train | | |
|---|---|---|---|---|---|---|---|
| | | PSNR (↑) | SSIM (↑) | LPIPS (↓) | PSNR (↑) | SSIM (↑) | LPIPS (↓) |
| 3DGS | – | 27.67 | 0.82 | 0.20 | 29.71 | 0.90 | 0.16 |
| Architecture | QNN → Vanilla MLP$_{same}$ | 27.71 | 0.83 | 0.20 | 29.70 | 0.89 | 0.16 |
| | QNN → Vanilla MLP (Mildenhall et al., 2020) | 27.74 | 0.83 | 0.20 | 29.72 | 0.89 | 0.16 |
| | QNN → Fourier MLP (Tancik et al., 2020) | 27.78 | 0.82 | 0.20 | 29.77 | 0.89 | 0.16 |
| | QNN → Hash-grid MLP (Müller et al., 2022) | 27.79 | 0.82 | 0.20 | 29.77 | 0.89 | 0.16 |
| | QNN → Erf MLP (Yang & Salman, 2019) | 27.68 | 0.82 | 0.20 | 29.66 | 0.89 | 0.16 |
| | QNN → Sinc MLP (Saratchandran et al., 2024) | 27.68 | 0.82 | 0.20 | 29.66 | 0.89 | 0.16 |
| | QNN → SIREN MLP (Sitzmann et al., 2020) | 14.49 | 0.62 | 0.51 | 14.66 | 0.67 | 0.49 |
| | QNN → FINER MLP (Liu et al., 2024) | 27.64 | 0.82 | 0.20 | 29.66 | 0.89 | 0.16 |
| | 2D Conv (3×3) → Linear (1×1) | 27.83 | 0.82 | 0.20 | 29.84 | 0.90 | 0.16 |
| Input | Queries ⊕ LF → Queries | 27.81 | 0.82 | 0.20 | 29.80 | 0.90 | 0.16 |
| | Queries ⊕ LF → LF | 27.61 | 0.82 | 0.20 | 29.96 | 0.90 | 0.16 |
| Queries | 2D → 3D location | 27.59 | 0.82 | 0.19 | 30.74 | 0.91 | 0.14 |
| | 2D → 3D location ⊕ direction | 27.66 | 0.82 | 0.20 | 29.87 | 0.90 | 0.16 |
| | 2D → Raymap (Mildenhall et al., 2020) | 27.73 | 0.82 | 0.20 | 29.85 | 0.90 | 0.16 |
| | 2D → Raymap ⊕ 2D | 27.88 | 0.82 | 0.20 | 30.03 | 0.90 | 0.16 |
| | 2D → Plücker (Plücker, 1828) | 27.67 | 0.82 | 0.20 | 29.85 | 0.90 | 0.16 |
| Activation | ReLU → Erf (Yang & Salman, 2019) | 26.43 | 0.80 | 0.22 | 27.98 | 0.87 | 0.18 |
| | ReLU → Sinc (Saratchandran et al., 2024) | 27.70 | 0.82 | 0.20 | 29.79 | 0.90 | 0.16 |
| | ReLU → FINER (Liu et al., 2024) | 27.33 | 0.81 | 0.22 | 28.62 | 0.87 | 0.19 |
| Encoding | Vanilla → Fourier (Tancik et al., 2020) | 27.85 | 0.82 | 0.20 | 29.90 | 0.90 | 0.16 |
| | Vanilla → Per-axis Fourier | 27.85 | 0.82 | 0.20 | 29.89 | 0.90 | 0.16 |
| | Vanilla → Exponential (Mildenhall et al., 2020) | 27.94 | 0.82 | 0.20 | 29.99 | 0.90 | 0.16 |
| 3DGS+QNN | – | 27.96 | 0.83 | 0.20 | 30.11 | 0.90 | 0.16 |

**Training Times**. 3DGS, 3DGS + QNN, MCMC, MCMC + QNN and Zip-NeRF models take 40, 50, 30, 43 and 270 minutes respectively for training on a Mip-NeRF 360 scene measured on a single V100 GPU. Thus, MCMC + QNN obtains Zip-NeRF quality without huge training times.

## 5.5 ABLATION STUDIES

To understand the core design choices of QNN, we conduct ablation studies on the NVS task, as it is the most complex and difficult. These experiments use the 3DGS baseline on all nine scenes of the Mip-NeRF 360 dataset (Barron et al., 2022) following Sec. 5.4. We report standard metrics on both the val and train sets to evaluate the generalization and fitting performance, respectively.

**Architecture.** We first ablate the network architecture, replacing the QNN with several MLP-based neural fields that take queries as inputs. This included a vanilla MLP (with same or different channels), a Fourier MLP, hash-grid MLP and MLPs with Sinc, Erf, SIREN or FINER activations. None of these alternatives improved NVS performance either on the val or the train sets in Tab. 4, which justifies our use of a convolutional architecture. Next, we also changed the 2D convolution to linear layer (3×3→1×1) in QNN. This result reinforces that convolution remains the key for learning high-frequency details.

**Input.** We next examined the input to the QNN. Tab. 4 findings show that using only queries or only low-frequency (LF) images decreases NVS performance on both the val and training sets. This confirms that concatenating both inputs is crucial for optimal results.

**Queries.** As explained in Sec. 4, we explored different queries, including:
- 3D location: Unprojecting pixel locations into 3D using camera parameters and splatted depth.
- 3D location ⊕ direction: Concatenating 3D view directions to the 3D locations.
- Raymap: Using 3D camera centers with view directions, as in NeRF (Mildenhall et al., 2020). Gao et al. (2024) refer to this construct as a raymap.
- Raymap ⊕ 2D: Appending the 2D queries to the raymap queries.
- Plücker: Using the Plücker rays (Plücker, 1828; Grossberg & Nayar, 2001) as queries.

Tab. 4 shows that 2D queries performed best on the validation set, while 3D location queries yielded the best performance on the training set. This divergence warrants further investigation, and we leave a deeper exploration of query performance for future work.

**Activation.** We then explored changing the ReLU activation within QNN to Sinc, Erf and FINER activations. These experiments use our proposed Qonvolution layer, with the only change being the activation. Tab. 4 shows that even these changes prove suboptimal to our best performance.

**Encoding.** QNN uses vanilla encodings. We also experiment with randomized Fourier (Tancik et al., 2020), per-axis Fourier and exponential style NeRF encodings (Mildenhall et al., 2020). Changing encodings from vanilla to any of these decreases performance on both val and train sets.

## 5.6 MORE EXPERIMENTS

**Scalability of QNN.** Fig. 4 shows the scalability of QNN with the MCMC baseline on the Mip-NeRF 360 dataset by varying the number of final gaussians. The figure confirms that adding QNN consistently improves the NVS task at varying number of gaussians.

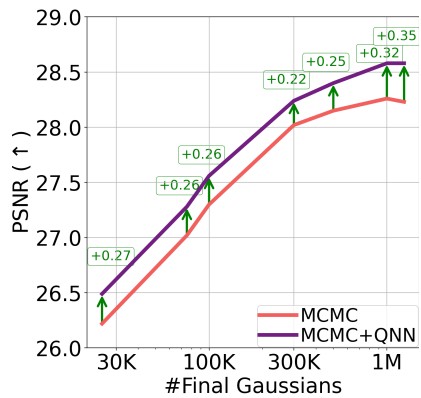

Figure 4: **Scalability of QNN.** Adding QNN consistently improves the NVS task for the MCMC baseline at varied number of final gaussians.

Table 5: **PSNR Edge Evaluation** for the NVS task on the Mip-NeRF 360 dataset. Adding QNN **improves** the $PSNR_{Edge}$ or the high-frequency details for both 3DGS and MCMC baselines. [Key: **Best**].

| Method | PSNR (↑) | $PSNR_{Edge}$ (↑) |
|---|---|---|
| 3DGS | 27.67 | 25.76 |
| 3DGS + QNN | **27.96** (+0.29) | **25.89** (+0.13) |
| MCMC | 28.26 | 26.28 |
| MCMC + QNN | **28.58** (+0.32) | **26.45** (+0.17) |

**Does QNN improve high-frequency details?** To quantitatively measure detail recovery, we followed the methodology of SASNet (Feng et al., 2025) and computed PSNR specifically on the image edges for the NVS task on the Mip-NeRF 360 dataset in Tab. 5. We calculate the $PSNR_{edge}$ metric by first identifying edge regions using the Canny edge detector (Canny, 1986) with thresholds 0, 100, followed by disk dilation of kernel size 3 (Feng et al., 2025). Tab. 5 results show that adding QNN consistently improves the $PSNR_{Edge}$ score for both the 3DGS and MCMC baselines. This quantitatively confirms that adding QNN improves the fidelity of high-frequency details (edges) in the outputs.

## 6 CONCLUSIONS

Accurately learning high-frequency signals is a challenge in computer vision and graphics, as neural networks often struggle with these signals due to spectral bias or optimization difficulties. While current techniques like Fourier encodings have made great strides in improving performance, there remains scope for improvement when presented with high-frequency information. This paper introduced Qonvolution Neural Network (QNN), a simple yet powerful modification using the neighborhood properties of convolution. Qonvolution convolves a low-frequency signal with queries to enhance the learning of intricate high-frequency signals. We empirically demonstrate that QNN enhance performance across a variety of high-frequency learning tasks crucial to both the computer vision and graphics communities, including 1D regression, 2D super resolution, 2D residual image regression, and NVS. In particular, by combining Gaussian splatting with QNN for NVS, we showcase state-of-the-art performance on real-world complex scenes, even outperforming powerful radiance field models on image quality. Future work involves exploring whether advances in neural implicit fields arising from orthogonal research directions are also beneficial for QNNs.

**Limitations.** QNNs require neighborhood information for processing. QNNs cannot be used in tasks without neighborhoods such as casting a single ray in a NeRF (Mildenhall et al., 2020) or in SDF (Park et al., 2019).

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
