# OpenReview forum: "Qonvolution: Towards Learning of High-Frequency Signals with Queried Convolution"
_ICLR.cc/2026/Conference — Submitted to ICLR 2026_

### Official Review · Reviewer_xHFh · 2025-10-27

**Soundness:** 3
**Presentation:** 2
**Contribution:** 3
**Rating:** 4
**Confidence:** 3

**Summary:**

The paper introduces Qonvolution / QNN (Queried Convolutional Neural Network), a convolutional module that jointly consumes (i) local neighborhoods of a low-frequency signal and (ii) per-pixel/per-sample query coordinates, and then predicts the missing high-frequency residual signal. Instead of fitting high-frequency content purely with coordinate MLPs or purely with CNNs over low-resolution images, QNN explicitly concatenates the encoded coordinates with the low-frequency approximation and applies convolution over local neighborhoods to regress the high-frequency component.

The authors evaluate QNN across four regimes: (1) 1D high-frequency regression, (2) 2D super-resolution, (3) 2D residual image regression, and (4) novel view synthesis (NVS). The paper also provides theoretical arguments that (a) adding neighborhood context and explicit query coordinates cannot worsen the optimal achievable risk compared to using either alone, and (b) high-frequency fidelity with pure Gaussian primitives may require exponentially many Gaussians, motivating a learned residual head instead of endlessly increasing primitive count.

**Strengths:**

- Unified viewpoint across tasks.
The same basic idea (concatenate query coordinates and a low-frequency approximation, then convolve locally to predict high-frequency residuals) is tested on 1D regression, 2D SR, and 2D residual regression, not just NVS. This reduces the risk that the method is a one-off trick.

- Lightweight and modular.
QNN is a shallow conv stack (e.g., 4 conv layers, 3×3 kernels, ~64 channels) trained jointly with the baseline renderer. It does not require ray marching, volumetric integration, or massive MLP decoders per ray, which keeps training time much closer to 3DGS/MCMC and far below Zip-NeRF.

- Some theoretical grounding.
The appendix offers two arguments:
(i) more contextual features (neighborhood + coordinates) can only improve the achievable optimal predictor’s error;
(ii) purely Gaussian splatting may require exponentially many primitives to drive MSE down, motivating a learned residual head instead of brute-force Gaussian proliferation.

**Weaknesses:**

- Lack of qualitative results.
There are many tasks for this paper, but only a few qualitative comparisons are provided by the author. It is hard for me to fully evaluate the performance of QNN based on the current results. Especially in NVS task, providing rendered videos will largely enhance this work.
- Effect size and statistical robustness.
Many gains in PSNR/SSIM on SR and NVS are in the +0.1–0.3 dB / +0.01 SSIM range. This is meaningful but not dramatic, so variance matters. The current draft does not clearly communicate per-scene standard deviations or multiple random seed evaluations.
- Novelty vs. existing coordinate-aware refinement nets.
The paper positions QNN as a new primitive. But conceptually, adding (x,y,...) coordinates as extra channels to a conv net and asking it to learn a residual correction on top of a coarse render is reminiscent of CoordConv-like ideas and of post-render refinement heads seen in prior SR / NeRF variants. The authors acknowledge that QNN reduces to known special cases (pure CNN if you drop queries, coordinate CNN if you drop some parts), which somewhat blurs how much is genuinely new versus systematized.
- Baseline fairness/training protocol transparency.
The NVS table mixes “Reported,” “Reproduced,” and “+QNN” numbers. It’s unclear to what extent the baselines (3DGS, MCMC) were retrained under exactly the same hyperparameters and optimization schedule as the QNN-augmented version, or whether the baselines were tuned as hard as QNN. This matters because a 0.2–0.4 dB PSNR bump could come from better optimization rather than architectural superiority.
- Definition of ‘low-frequency’ vs ‘high-frequency’.
The paper repeatedly relies on this split (e.g., “low-frequency splatted image from 3DGS” vs. “high-frequency residual predicted by QNN”), but the formalization is scattered across sections and tasks. A more consistent, quantitative definition (e.g., specific band-limited cutoff or residual construction) would make the paper easier to follow, and would help future researchers reproduce the same decomposition.

**Questions:**

These are the major questions that will affect my ratings:
- Qualitative results.
Can you provide more qualitative results for each task? If there is a way to provide video during rebuttal, the results of NVS will be more convincing.
- Variance / statistical significance.
For Tables 1–3 and the NVS table: please report per-scene or per-seed variance. Are the ~0.2–0.4 dB PSNR gains statistically consistent across scenes, or dominated by a few?
- High-frequency decomposition.
In NVS you describe the QNN as predicting a high-frequency residual that gets added back to the splatted image. How exactly is that residual defined across datasets and tasks? Is it literally GT - 3DGS_render per pixel? If so, is there any regularization to keep QNN from hallucinating view-inconsistent detail across viewpoints?

---

> ### Author Response · Authors · 2025-11-20
> **Response to R4 (xHFh) [1/2]**
>
> We thank the reviewer for the feedback. We first answer the questions in the Questions section followed by the ones listed in the Weaknesses section.
>
> ## R4Q1: Qualitative Video results, especially for NVS
>
> We added an `index.html` page in the supplementary material to view and compare rendered NVS video results of the Mip-NeRF 360 dataset. Please unzip the supplementary, navigate to the unzipped directory and follow README.md to see the qualitative video results. These rendered results show that adding QNN, faithfully reconstructs the scene providing higher quality renderings compared to the 3DGS baseline.
>
> ## R4Q2: Variance, Statistical Significance
>
> **NVS With 3DGS Baseline on Mip-NeRF 360 dataset**
>
> Method | Seed | | Val| | | Train| |
> -------|-|-|-|-|-|-|-|
> .    |     |PSNR   |SSIM  |LPIPS | PSNR  | SSIM | LPIPS|
> .    | 222 | 27.67 | 0.82 | 0.20 | 29.71 | 0.90 | 0.16 |
> 3DGS | 111 | 27.65 | 0.82 | 0.20 | 29.69 | 0.90 | 0.16 |
> .    | 555 | 27.72 | 0.82 | 0.20 | 29.69 | 0.90 | 0.16 |
> Mean $\pm~\sigma$ | -   | **27.68 $\pm$ 0.03** | 0.82 $\pm$ 0.00 | 0.20 $\pm$ 0.00 | **29.69 $\pm$ 0.02** | 0.90 $\pm$ 0.00 | 0.16 $\pm$ 0.00 |
> ||||||||
> .| 222 | 27.96 | 0.83 | 0.20 | 30.11 | 0.90 | 0.16 |
> 3DGS + QNN | 111 | 27.92 | 0.83 | 0.20 | 30.12 | 0.90 | 0.16 |
> . | 555 | 27.93 | 0.83 | 0.20 | 30.12 | 0.90 | 0.16 |
> Mean $\pm~\sigma$ | -   | **27.93 $\pm$ 0.02** | 0.83 $\pm$ 0.00 | 0.20 $\pm$ 0.00 | **30.12 $\pm$ 0.01** | 0.90 $\pm$ 0.00 | 0.16 $\pm$ 0.00 |
>
> The table above shows that integrating QNN into the 3DGS baseline for NVS on the Mip-NeRF 360 dataset yields statistically significant improvements.
>
> **NVS with MCMC Baseline on Mip-NeRF 360 dataset**
>
> Method | Seed | | Val| | | Train| |
> -------|-|-|-|-|-|-|-|
> .    |     |PSNR   |SSIM  |LPIPS | PSNR  | SSIM | LPIPS|
> .    | 222 | 28.26 | 0.84 | 0.17 | 30.31 | 0.91 | 0.14 |
> MCMC | 111 | 28.22 | 0.84 | 0.17 | 30.32 | 0.91 | 0.14 |
> .    | 555 | 28.31 | 0.84 | 0.17 | 30.31 | 0.91 | 0.14 |
> Mean $\pm~\sigma$ | -   | **28.26 $\pm$ 0.03** | 0.84 $\pm$ 0.00 | 0.17 $\pm$ 0.00 | **30.31 $\pm$ 0.02** | 0.91 $\pm$ 0.00 | 0.14 $\pm$ 0.00 |
> ||||||||
> .| 222 | 28.58 | 0.84 | 0.17 | 30.69 | 0.91 | 0.14 |
> MCMC + QNN | 111 | 28.53 | 0.84 | 0.17 | 30.67 | 0.91 | 0.14 |
> . | 555 | 28.54 | 0.84 | 0.17 | 30.69 | 0.91 | 0.14 |
> Mean $\pm~\sigma$ | -   | **28.55 $\pm$ 0.02** | 0.84 $\pm$ 0.00 | 0.17 $\pm$ 0.00 | **30.68 $\pm$ 0.01** | 0.91 $\pm$ 0.00 | 0.14 $\pm$ 0.00 |
>
> The table above shows that integrating QNN into the MCMC baseline for NVS on the Mip-NeRF 360 dataset yields statistically significant improvements. We added these results as Tab. 11 in the supplementary.
>
> ## R4Q3: Are PSNR gains statistically consistent across scenes, or dominated by a few?
>
> Method | Mean | Mean Outdoor | Mean Indoor | garden	| bicycle | stump | bonsai | counter | kitchen | room | treehill | flowers
> -------|-|-|-|-|-|-|-|-|-|-|-|-|
> 3DGS | 27.68 | 24.97 | 31.06 | 27.66 | 25.61  | 26.89  | 32.16  | 29.10  | 31.40 | 31.59  | 22.82 | 21.88 |
> 3DGS + QNN | **27.93** | **25.08** | **31.50** | 27.89 |	25.66 |	27.17 | 32.57 | 29.23 |	31.70 |	32.51 |	22.73 | 21.97
> MCMC | 28.26 | 25.38 | 31.96 | 28.00 | 25.92 | 27.22 | 33.25 | 29.86 | 32.46 | 32.26 | 23.33 |	22.24
> MCMC + QNN | **28.55** | **25.43** | **32.44** | 28.24 | 25.97 | 27.53 | 33.72 | 29.96 | 32.74 | 33.34	| 23.25 | 22.17
>
> We report the mean PSNR across three different seeds with both 3DGS and MCMC baselines for NVS on the Mip-NeRF 360 dataset in the above table. The PSNR gains are not uniformly distributed across all scenes and larger improvements are observed specifically in indoor scenes. We added these results as Tab. 12 in the supplementary.
>
> ## R4Q4: High-frequency decomposition: How exactly is that residual defined across datasets and tasks?
>
> Residual/HF | Output | Loss Function
> ----|-| -|
> QNN(q, LF) | LF + QNN(q, LF) | Loss[LF + QNN(q, LF), GT]
>
> That's a great question. The QNN consistently defines the high-frequency residual across various tasks and datasets. As shown in the table, for all tasks, the QNN (denoted as $QNN(q, LF)$) predicts a residual that is added to the low-frequency input ($LF$) to produce the final output, which is then evaluated against the ground truth ($GT$) using a task-specific loss function:
>
> Task | Loss | Low-Freq Input | Queries (q) | Convolution
> --- | - | - | - | - |
> 1D Reg | $L_1$ | Oracle LF | 1D coordinates | 1D
> 2D SR  | $L_2$ | Oracle LF | 2D coordinates | 2D
> 2D Reg | $L_1$ | 3DGS Predictions | 2D coordinates | 2D
> 3D NVS | 0.8$L_1$ + 0.2$(1-SSIM)$ | 3DGS Predictions | 2D coordinates | 2D

---

> ### Author Response · Authors · 2025-11-20
> **Response to R4 (xHFh) [2/2]**
>
> ## R4Q5: Is HF literally GT - 3DGS_render per pixel? If so, is there any regularization to keep QNN from hallucinating view-inconsistent detail across viewpoints?
>
> Yes, in the NVS task, the residual is indeed defined as the per-pixel difference between the ground truth (GT) and the 3DGS render. However, the QNN does **not regress this residual** separately; instead, the splatted image is added to the QNN predictions, and this combined output is then regressed against the GT, as detailed in the response to R4Q4.
>
> There is **no explicit regularization** implemented to prevent the QNN from hallucinating view-inconsistent details across different viewpoints.
>
>
> ## R4Q6: Novelty wrt coordinate CNNs.
>
> . | CoordCNN | QNN |
> -- | - | - |
> Queries | 2D coordinates | 1D, 2D, 3D coordinates, raymap, plucker or even their concatenation |
> Claim | Spatial coordinate awareness | High-Frequency Learning
> Theory | ╳ | $\checkmark$
> Tasks | ImageNet classification, MNIST detection, LSUN GAN, RL | 2D SR, 3D NVS
>
> QNNs represent a substantial generalization over CoordCNNs, underpinned by a distinct theoretical foundation and motivation. QNNs learn the high-frequency residual signal, which achieves high-fidelity results in tasks like NVS, significantly extending the scope and impact beyond traditional CoordCNN applications.
>
> ## R4Q7: The NVS table mixes Reported, Reproduced, and +QNN numbers.
>
> We disagree. The NVS table distinguishes between "Reported," "Reproduced," and "+QNN" numbers, which are defined as follows: "Reported" refers to figures directly quoted from the original papers, "Reproduced" indicates results obtained by running the codebase independently, and "+QNN" denotes the performance achieved when the codebase is augmented with QNN.
>
> ## R4Q8: Training protocol transparency: Are baselines retrained under the same hyperparameters and optimization schedule as the QNN-augmented version, or whether the baselines were tuned as hard as QNN.
>
> Yes, all baselines and their QNN-augmented versions are trained using identical protocols, including the same number of iterations, loss functions, hyperparameters, and optimization schedules. Notably, the reproduced baseline results in Tab. 3 **surpass the reported numbers** from their original papers.
>
> ## R4Q9: Definition of ‘low-frequency’ vs ‘high-frequency’. More consistent, quantitative definition (e.g., specific band-limited cutoff or residual construction)
>
> We define low and high-frequency components through a decomposition of the original function $f$. The low-frequency component, $f_{low}$, is derived by convolving $f$ with a low-pass filter, denoted as $h_{LPF}$. The high-frequency component, $f_{high}$, is then simply the residual obtained by subtracting $f_{low}$ from $f$. This relationship can be expressed as $f = f_{low} + f_{high} = (f * h_{LPF}) + f_{high}$.
>
> As an example, in our 1D regression experiment, we set the low pass filter's normalized cutoff to $0.125$.

---

> ### Comment · Reviewer_xHFh · 2025-11-24
>
> Thank you for the detailed rebuttal. Most of my original concerns have been addressed, and I appreciate the additional experiments and clarifications. However, I still have two points where I would like further explanation. If these can be clarified convincingly, I am inclined to increase my rating.
> - R4Q6 – Novelty and generality of QNN.
> In the rebuttal and paper, you claim that QNN can query using 1D, 2D, 3D coordinates, ray maps, Plücker coordinates, or even their concatenation. However, in the actual experimental setup, even for the 3D tasks, the queries appear to be 2D image coordinates (plus low-frequency images), rather than truly 3D/ray/Plücker queries. This makes the claim feel somewhat over-stated and not clearly beyond what CoordConv-style architectures already support. Could you please clarify in which experiments you actually use 3D coordinates, ray maps, or Plücker coordinates as queries?
>
> - R4Q5 – Cross-view consistency with 2D post-processing. For the NVS setting, QNN is applied as a per-view 2D refinement of Gaussian predictions. Without any explicit cross-view consistency constraint, it is not clear to me how this post-processing avoids degrading 3D consistency (e.g., producing view-dependent hallucinations that are not geometrically compatible across views). Could you elaborate on why this does not harm 3D consistency in practice?

---

> ### Author Response · Authors · 2025-11-24
> **Response to R4 xHFh**
>
> Thank you R4 for your comments.
>
> ## Point1: Which experiments use 3D coordinates, ray maps or Plücker coordinates?
>  Queries | | Val| | | Train| |
> -------|-|-|-|-|-|-|
> .|PSNR|SSIM|LPIPS|PSNR|SSIM|LPIPS|
> 3D Location | 27.59 | 0.82 | 0.19 | **30.74** | **0.91** | **0.14** |
> 3D Location + Direction | 27.66 | 0.82 | 0.20 | 29.87 | 0.90 | 0.16 |
> Raymap | 27.73 | 0.82 | 0.20 | 29.85 | 0.90 | 0.16 |
> Raymap + 2D | 27.88 | 0.82 | 0.20 | 30.03 | 0.90 | 0.16 |
> Plücker | 27.67 | 0.82 | 0.20 | 29.85 | 0.90 | 0.16 |
> **2D (Ours)** | **27.96** | **0.83** | 0.20 | 30.11 | 0.90 | 0.16 |
>
> NVS experiments: Tab. 4 (Queries) and Sec 4.4 (Queries) report results with different queries on the Mip-NeRF 360 dataset. We added an experiment with Plucker coordinates as well. As shown in the table above (a breakdown of Tab. 4), 2D queries performed best on the validation set, while 3D location queries yielded the best performance on the training set.
>
> ## Point2: Why adding QNN does not harm 3D consistency in practice?
>
> We believe QNN does not harm 3D consistency in practice due to the following reasons:
>
> - **Residual Framework over 3DGS**: The QNN predictions are added as a residual refinement on top of the 3D Gaussian Splatting (3DGS) output. Since 3DGS provides a good, view-consistent starting point, QNN only models the high-frequency error, benefiting from the consistency of the base signal.
> - **Strong Supervision**: The overall model output (3DGS + QNN residual) is strictly supervised by Ground Truth (GT) images identical to the baseline 3DGS, which anchors the refinement process to consistency.
> - **Interpolation Regime**: The NVS task operates in an interpolation regime, where test views are distributed similarly to training views. Neural networks like QNN are good at interpolation, which helps maintain consistency between views.
>
> We hope these clarifications fully address your concerns. We are happy to answer or clarify any further questions you may have. And should the rebuttal and the paper resolve your concerns, we would greatly appreciate if you could increase the rating for our paper.

---

> ### Comment · Reviewer_xHFh · 2025-11-24
>
> Although I still have some concerns about the 3D consistency part, the authors have addressed most of my major concerns, and I will raise my rating to 6.

---

> ### Author Response · Authors · 2025-11-24
> **Thank you R4 (xHFh)**
>
> Thank you R4 (xHFh) for raising the score of our paper to 6.

---

### Official Review · Reviewer_Vfyj · 2025-10-30

**Soundness:** 3
**Presentation:** 2
**Contribution:** 2
**Rating:** 4
**Confidence:** 4

**Summary:**

The paper proposes queried convolutional neural networks (QNNs) as a signal representation analogous to implicit neural representations (INRs), but designed to improve ability to fit high frequency signals (ie mitigate spectral bias). The architecture is similar to a standard CNN, except that the QNN takes as input not only an image but also its encoded queries (pixel coordinates). Experiments apply QNNs to 1D and 2D regression, 2D super-resolution, and novel view synthesis.

**Strengths:**

The main idea of augmenting an INR with low-frequency image features, or augmenting a CNN with coordinate features, makes sense as a way to combine the strengths of CNNs and INRs. The discussion of related work is clear and fairly complete (with one exception noted as a question). The quantitative results in table 2 show large PSNR improvements by adding QNN. I also appreciate the inclusion of a theoretical result relating the expressive capacity of QNNs and CNNs. The novel view synthesis experiments use a large suite of datasets and baseline methods, and training times are reported. I also appreciate the honest mention of a key limitation (the requirement of neighborhood information) in the conclusions.

**Weaknesses:**

Many of these weaknesses should be read as suggestions to improve the paper presentation. Some are weaknesses with respect to the motivation and experimental setup. Listed in rough order of appearance, not by importance.
- Line 042 cites Rahimi & Recht, 2007, in the context of neural networks for processing 1D data. This strikes me as a bit misleading, since that paper does not involve neural networks.
- Figure 1, line 87, and line 140 mention a combination of the proposed Queried CNN (QNN) with 3DGS, but since these two methods use completely different representations (and since QNN operates in 2D while 3DGS operates in 3D) it is not clear at these points in the paper how they would be combined.
- Line 124 suggests that implicit representations of images are a key aspect of diffusion-model-based super-resolution. However, the two papers cited in this line do not involve diffusion models. It is true that both diffusion models and implicit representations can be used for super-resolution, but the current framing makes these approaches seem more intertwined than they are (or at least, than they are in the papers cited).
- The sentence on line 129 is also a bit misleading. It makes it seem like the continuous nature of the NeRF representation is what enables its high fidelity. However, a few sentences later the paper notes (correctly) that 3DGS works just fine at the same task; since 3DGS uses a discrete representation, it is surely not the continuity of NeRF’s representation that explains its performance.
- From section 4.2 and Theorem 1 it seems that part of the paper’s contribution is a theoretical investigation of the predictive power of QNNs compared to MLPs and CNNs. This should be mentioned as a contribution in the bullet point list at the end of the introduction, which as written left me surprised to see any theory in the paper.
- Some more specific comments on Theorem 1: (i) I’m not sure why feature map 1 is described as an MLP. That description makes it sound like a standard INR, but the actual definition of feature map 1 is just some approximation of the true function, not including the coordinates as input. It would be preferable to include some common INR in the theorem statement, and if that is not feasible then I suggest renaming feature map 1 from MLP to just “approximation” or something along those lines, as there is not necessarily an MLP involved in f^low. (ii) Line 248 introduces the theorem saying that “we perfectly approximate the target function”--this strikes me as a bit of an overstatement. The theorem states that the minimum mean squared error achievable by a QNN is zero, but as QNN optimization is still nonconvex there is no guarantee that a QNN will actually achieve this minimum error. (iii) The theorem focuses on the case of 1D input and 1D output; that should be noted when the theorem is introduced, not just in the theorem statement.
- Line 206-208 argues that QNN’s structure of retaining the spatial dimensions of the input and output is an advantage over typical implicit neural representations (INR), because it avoids the need for reshaping. This argument does not make much sense to me, because reshaping is a fairly efficient operation, and there are clear benefits to operating on individual coordinates because (i) it means an INR can be used in an inverse problem where each measurement involves only a portion of the signal, without evaluating the entire signal for each measurement update, and (ii) it means an INR can be used to represent a signal in arbitrary dimensions, with negligible architectural changes. Perhaps a more convincing argument for this property of QNNs is that the neural network need only be evaluated once for the entire signal, rather than once per coordinate. But it does come at some cost in terms of flexibility, since it’s not obvious how a QNN would work for a 3D signal.
- The 2D super-resolution results in Table 1 show very marginal improvement over baselines. It’s not clear if the model sizes and training times were comparable between methods, so I can’t be confident that the small improvement is really due to the architectural change of using QNN.
- I’m not sure I understand Figure 3. The “GT LP” signal is supposed to be a dashed line, but I don’t see a dashed line in the figure. I suspect it is because it is covered up by the predictions of the MLP, CNN, and QNN, which all seem to predict a fairly smooth function. If this is the case, it would seem that these methods are all basically learning the low-pass portion of the target, which doesn’t really support the case that QNN is mitigating spectral bias more than the other methods.
- It is a bit confusing that the 2D super-resolution experiments train on a dataset of images, whereas all the other experiments are in the INR setting where the network is trained individually on each signal. The reasoning for including both types of experiments needs to be explained; currently even the fact that these experiments are of different types is not very obvious (I only noticed because of the description of the training data in section 5.2, and the footnote on page 2).
- The wording for the dataset description around line 350 is not clear, specifically the term “residual image.” I think this is saying that the QNN takes a 3DGS splatted image (and its coordinates) as input, and the output is compared against a mip-NeRF rendered image. But the wording sounds like the target is a residual image? I am also confused more broadly by the setup of this experiment, which seems to treat 3DGS images as a low-frequency approximation of Mip-NeRF images…usually 3DGS produces more detailed images than Mip-NeRF, so this seems backwards to me. Also since the QNN operates on images, using it as a postprocessing step (which I think is what is proposed) could break the 3D consistency that both 3DGS and Mip-NeRF enforce.
- As much as I appreciate the extensiveness of the experiments reflected in table 3 and table 4, and that the QNN was trained alongside the main radiance field to encourage 3D consistency (unlike in table 2), the quantitative results themselves show only marginal improvement (less than 0.5 db of PSNR) by adding QNN.

**Questions:**

This is neither a strength, weakness, nor question, so I am listing it here. The main idea of the method bears some similarity to part of the patch-based diffusion method proposed in https://arxiv.org/abs/2406.02462. While the goal and architectures differ between that paper and this one, the idea of concatenating the image pixel coordinates to the image itself as input to a neural network, is used in that paper. It is sufficiently different as to not impinge on the novelty of this work, but I suggest mentioning it to acknowledge that idea.

There are a few questions embedded in the “weaknesses” section.

---

> ### Author Response · Authors · 2025-11-20
> **Response to R3 (Vfyj)**
>
> We thank the reviewer for the feedback. We first answer the questions in the Questions section followed by the ones listed in the Weaknesses section.
>
> ## R3Q1: Cite PaDIS paper.
>
> Thanks for suggesting this work. We have added this paper to the Sec. 2: NVS subsection.
>
> ## R3Q2: Why cite Rahimi and Recht ?
>
> This paper was the precursor for Fourier encodings for 1D signal, and so we included it. We fixed this.
>
> ## R3Q3:  Fig. 1, L87, L140 combine QNN with 3DGS. How are they combined?
>
> We add queries to the splatted 3DGS images (which are in 2D). Specifically, the 2D output image produced by 3DGS serves as the low-frequency signal, and the corresponding 2D queries are concatenated to it. This concatenated input is then processed by the QNN to predict the high-frequency residual correction. Please see Fig. 7 in the supplementary for the figure.
>
> ## R3Q4: L124 suggests that implicit representations of images are a key aspect of diffusion-model-based super-resolution. However, the two papers cited in this line do not involve diffusion models.
>
> L124 focuses on the use of implicit representations of images as a key aspect of certain super-resolution methods. We cited those two papers because they exemplify the use of implicit representations in image restoration tasks, not because they are based on diffusion models.
>
> ## R3Q5: L129 Misleading: Continuous nature of the NeRF representation is what enables its high fidelity.
>
> We corrected it.
>
> ## R3Q6: Add Theorem 1 to the bulleted contribution
>
> Thank you! We added.
>
> ## R3Q7: Why feature map as MLP?
>
> Thank you for pointing this out. We have revised the notation to MLP(LF) to explicitly indicate that the input to MLP is the low-frequency approximation signal.
>
> ## R3Q8: L248: "we perfectly approximate the target function” is an overstatement.
>
> We confirm that this statement holds true only under the idealized assumptions (unlimited data and computation) set forth in our theoretical justification. We have added a clarification to the manuscript to explicitly state this context and prevent future misinterpretation.
>
> ## R3Q9: Say Theorem 1 is for 1D input and 1D output.
>
> Thank you for your suggestion. We would like to clarify that Theorem 1 already allows for **multi-dimensional inputs**. This is indicated by the description of the input space $\Omega \subset \mathbb{R}^d$ at the beginning of the theorem statement, where $d$ can be greater than one. To ensure clarity regarding the output, we have updated the manuscript to explicitly state that the theorem considers real-valued functions, i.e., **1D output**.
>
> ## R3Q10: L206-208 Reshaping argument does not make much sense to me. A more convincing argument is that the neural network need only be evaluated once for the entire signal, rather than once per coordinate.
>
> Thank you. We added this.
>
>
> ## R3Q11: Small improvement in Tab. 1 is due to architectural changes.
>
> Yes, it is due to architectural changes.
>
> ## R3Q12: Fig. 3 MLP, CNN, and QNN, predict a fairly smooth function. If this is the case, it would seem that these methods are all basically learning the low-pass portion of the target, which doesn’t really support the case that QNN is mitigating spectral bias more than the other methods.
>
> The PSNR subfigure (on the right of Fig. 3) quantitatively evaluates the performance on this 1D regression task, confirms that QNN does improve fidelity compared to the MLP and CNN baselines. While the magnitude of improvement may be small for this specific 1D task, this quantitative gain still supports our claim that QNN is more effectively mitigating spectral bias.
>
> ## R3Q13: Experiments of different types are not very obvious.
>
> Thank you. We added a clarifying line in Sec. 4 (Experiments) to make the distinction between the different task categories more explicit for the reader.
>
> ## R3Q14: L350 (Sec. 5.3) QNN output compared against a Mip-NeRF rendered image?
>
> We apologize for the confusion arising from the mention of "Mip-NeRF image" when referring to the dataset. We confirm that we are using images from the **Mip-NeRF 360 dataset**.
>
> The final output for the 2D image regression task is **not compared** against the rendered image from the **Mip-NeRF method**; it is compared against the **GT image from the Mip-NeRF 360 dataset**. Specifically, the overall prediction is calculated as the 3DGS splatted image (low-frequency baseline) plus the QNN's predicted residual image (high-frequency correction). The loss is then computed between this final summed prediction and the GT Mip-NeRF 360 dataset image. We added this clarification to Sec. 5.3.
>
> ## R3Q15: Tab. 4 show marginal improvements of 0.5 PSNR.
>
> The improvements in Tab. 4, are actually substantial in generative modeling because PSNR is a logarithmic metric. Further analysis in R4Q2 confirms the statistical significance of this improvement through a variance analysis.

---

> > ### Comment · Reviewer_Vfyj · 2025-11-23
> >
> > Thanks for clarifying many of my questions and making revisions in the manuscript. A few questions remain.
> > 1. My question about the source of the small improvements in Table 1 was not really addressed. What evidence supports that these small improvements are due to the architectural change, rather than e.g. differences in model size or training time? If the model sizes and training times are equal across models (or even if they are unequal but QNN is not given more time or memory than the other methods), then it would be more clear that the improvement is due to the architecture.
> > 2. I appreciate the addition of the edge-focused experiment in Figure 4 and Table 5, since it gets at the question of addressing spectral bias through evaluation of improvement in edges (which are high-frequency). That said, the results themselves show a larger improvement in metrics at the full-image level, and a comparatively smaller improvement when evaluation is restricted to edges. If the main benefits of QNN are in addressing spectral bias, I would expect the majority of the improvement to be concentrated near edges, which is not the case. Do the authors have any idea why this is?
> > 3. It seems that the use of QNN as fitting a residual image was not very clear originally, and I appreciate that the authors have tried to clarify that in the revision. I think part of this confusion might have stemmed from Figure 2, which shows QNN producing a high-frequency image as its output, rather than a residual image. I suggest modifying Figure 2 to be as accurate as possible, to avoid future reader confusion.

---

> > > ### Author Response · Authors · 2025-11-24
> > > **Response to R3 Vfyj Comment**
> > >
> > > We thank the reviewer for the constructive comments.
> > >
> > > ## 1. What evidence supports that improvements are due to architectural change in Tab. 1
> > >
> > > The observed improvements are directly attributable to the architectural changes introduced by QNN.
> > > In all comparative experiments, we ensured that training time, and other hyperparameters remained constant between the baseline and the QNN-integrated model. Since all other factors are controlled, the consistent performance gains are a consequence of the QNN architecture.
> > >
> > > ## 2. Tab. 5 results show comparatively smaller improvement at edges
> > >
> > > We acknowledge that the relative improvement on the edge PSNR in Tab. 5 may appear smaller than the overall PSNR gain. However, we emphasize two points:
> > > - **Consistent Improvement**: The important result is that QNN consistently improves performance on this challenging, high-frequency-focused metric for both baselines.
> > > - **Logarithmic Scale**: Since PSNR is a logarithmic metric (in decibels), even small numerical increases represent a substantial reduction in the mean squared error (MSE) and, therefore, a significant improvement in image fidelity.
> > >
> > > ## 3. Modify Fig. 2
> > > Thank you for the suggestion; we modified Fig. 2 in the updated manuscript for improved clarity.
> > >
> > > We hope these clarifications fully address your concerns. We are happy to answer or clarify any further questions you may have. And should the rebuttal and the paper resolve your concerns, we would greatly appreciate if you could increase the score for our paper.

---

> > > > ### Comment · Reviewer_Vfyj · 2025-11-24
> > > >
> > > > Regarding point 1, was model size (number of trainable parameters) also controlled, or does adding QNN come with an increase in model size?
> > > >
> > > > Regarding point 2, I realize PSNR is logarithmic and there is still an improvement at the edges. However, the improvement at the edges is still smaller than the improvement overall, so I would still appreciate some refinement of the claim that QNN is working by reducing spectral bias, since if that were the case we would expect a larger improvement at the edges compared to smooth regions. We see that QNN is working better overall and that includes improvement at edges, but the improvement is not concentrated near edges so I would say QNN is effective overall but not necessarily specifically at mitigating spectral bias.
> > > >
> > > > Regarding point 3, I see the modification to figure 2 but I'm a bit confused by it. The revised figure makes it look like QNN outputs a high-resolution image that is grayscale or with some other color shift, which is then added to a low-resolution input image to get the final high-resolution image. Is this really what the QNN output looks like? I was expecting it to look more like a high-frequency residual image (similar to an edge map). Is my expectation incorrect, or is this perhaps a stylized image instead of a real example of QNN output?

---

> > > > > ### Author Response · Authors · 2025-11-24
> > > > > **Response to Followup R3 Vfyj**
> > > > >
> > > > > Thank you, once again, R3, for your quick and thoughtful follow-up questions.
> > > > >
> > > > > ## Point1: Was model size (number of trainable parameters) also controlled ?
> > > > >
> > > > > Params = Trainable Params | CNN (Baseline) | QNN (Proposed) | Increase (%)
> > > > > ---- | - | - | - |
> > > > > Generator | 16,697,987 | 16,699,139 | +0.0068
> > > > > Discriminator | 4,376,897 |  4,376,897 |  +0.0000
> > > > >
> > > > > We report the parameters and trainable parameters of the two models for the Real-ESRGAN experiments. The number of trainable parameters remained nearly identical, ensuring a fair comparison based on architectural effectiveness. This increase in params happens because input to the QNN is a 5-channel image (3 RGB + 2 coordinates) compared to 3-channel RGB image as input to the CNN. The minimal increase confirms that the observed performance gains are not because of increase in model capacity.
> > > > >
> > > > > ## Point2: Refine Spectral Bias Claim
> > > > >
> > > > > Thank you for suggesting. We have updated the claim regarding spectral bias in the bulleted list on Page 2 of the manuscript to ensure greater precision and avoid overclaim.
> > > > >
> > > > > ## Point3: Fig. 2 Update
> > > > > We replaced the colored image with the High-Frequency (HF) image in Fig. 2, as requested. Please check the updated manuscript.
> > > > >
> > > > > Again, we are happy to answer or clarify any further questions you may have.

---

> > > > > > ### Comment · Reviewer_Vfyj · 2025-11-24
> > > > > >
> > > > > > Thanks for making these changes; I am raising my score to 6. Please add the information about approximate equivalence between model size and training time to the final paper.

---

> ### Author Response · Authors · 2025-11-24
> **Thank you R3 (Vfyj)**
>
> Thank you R3 (Vfyj) for raising the score of our paper to 6. Following your suggestion, we added the model size and training time clarifications with increase in the number of params to Sec. 5.2 Implementation paragraph.

---

### Official Review · Reviewer_cc5r · 2025-11-01

**Soundness:** 3
**Presentation:** 3
**Contribution:** 2
**Rating:** 4
**Confidence:** 4

**Summary:**

The paper proposes Qonvolution Neural Networks (QNNs) — a new way to learn high-frequency signals by combining low-frequency image features with coordinate queries through a convolution operation. This approach aims to overcome spectral bias in neural networks. The authors test QNNs across multiple domains: 1D regression, image super-resolution, residual image regression, and novel view synthesis (NVS), showing consistent improvements. Notably, when paired with 3D Gaussian Splatting, QNNs outperform Zip-NeRF in image quality while training much faster.

**Strengths:**

QNN is a neat, intuitive idea that merges the spatial inductive bias of CNNs with the flexibility of coordinate-based models. The method is simple to implement but demonstrates strong gains, especially for challenging 3D view synthesis. The theoretical result provides a clear rationale for why adding neighborhood and query information helps. Experiments are broad and well executed, with clear tables and ablations that show real benefits from each design choice. The NVS results are particularly impressive, suggesting QNNs can close the quality gap with far heavier NeRF-based systems.

**Weaknesses:**

While effective, the idea feels like an incremental step over CoordConv or query-conditioned CNNs. The theoretical justification is elegant but based on idealized assumptions. The gains in simpler 1D or 2D tasks are quite small, and the role of QNN in complex architectures like SR3 isn’t deeply analyzed. The method also depends on access to low-frequency signals, limiting generalization to cases without them. Lastly, there’s little discussion of computational overhead or scalability throughout the paper.

**Questions:**

- How is QNN fundamentally different from CoordConv beyond combining with low-frequency inputs?

- How sensitive are results to the quality of the low-frequency signal?

- Does QNN actually recover more fine-grained details, or mainly improve smoothness and consistency in outputs?

---

> ### Author Response · Authors · 2025-11-20
> **Response to R2 (cc5r) [1/2]**
>
> We thank the reviewer for the feedback. We first answer the questions in the Questions section followed by the ones listed in the Weaknesses section.
>
> ## R2Q1: Incremental Idea. Difference between CoordCNN and QNN.
>
> . | CoordCNN | QNN |
> -- | - | - |
> Queries | 2D coordinates | 1D, 2D, 3D coordinates, raymap, plucker or even their concatenation |
> **Core Claim** | Spatial coordinate awareness | High-Frequency Learning
> Theory | ╳ | $\checkmark$
> Tasks | ImageNet classification, MNIST detection, LSUN GAN, RL | 2D SR, 3D NVS
>
> QNNs represent a substantial generalization over CoordCNNs, underpinned by a distinct theoretical foundation and motivation. QNNs learn the high-frequency residual signal, which achieves high-fidelity results in tasks like NVS, significantly extending the scope and impact beyond traditional CoordCNN applications.
>
> ## R2Q2: Sensitivity to the low-frequency signal quality.
>
> This is a fantastic question! We used the final number of Gaussians in the MCMC baseline on Mip-NeRF 360 dataset as a proxy for the quality of the low-frequency signal. As shown in Fig. 4 (in the updated paper, referenced in R2Q9), the results confirm that adding QNN provides consistent performance improvements (in terms of PSNR) across the entire spectrum of varying low-frequency signal qualities (i.e., varying number of Gaussians). This demonstrates the robustness of QNN, affirming its ability to **effectively model the high-frequency signal even when the low-frequency input quality is sub-optimal**.
>
>
> ## R2Q3: Does QNN actually recover more fine-grained details, or mainly improve smoothness and consistency in outputs?
>
> Method | Mean $\text{PSNR}$ | Mean $\text{PSNR}_{\text{edge}}$
> -------|-|-|
> 3DGS | 27.68 | 25.76
> 3DGS + QNN | **27.96** | **25.89**
> MCMC | 28.26 | 26.28
> MCMC + QNN | **28.55** | **26.45**
>
> We address the question of whether QNN recovers fine-grained details versus merely improving smoothness. To quantitatively measure detail recovery, we followed the methodology of SASNet [A] and computed PSNR specifically on the image edges for the NVS task on the Mip-NeRF 360 dataset. We calculate the $\text{PSNR}_{\text{edge}}$ metric by first identifying edge regions using the Canny edge detector followed by disk dilation [A]. The results show that **adding QNN consistently improves the $\text{PSNR}_{\text{edge}}$ score for both the 3DGS and MCMC baselines**. This quantitative evidence confirms that QNN improves the fidelity of fine-grained details (edges) in the outputs.
>
> Reference:
>
> - [A] SASNet: Spatially-Adaptive Sinusoidal Neural Networks, Feng et al., ArXiv 2025
>
> ## R2Q4: Theoretical justification is elegant but based on idealized assumptions.
>
> We agree that the theoretical justification is based on an idealized setting (unlimited by data, or computation). However, our result is highly insightful, because it formally demonstrates that for accurate approximation of the target function, the network input must explicitly include the queries and the convolution.
>
> Furthermore, Corollary 2.1 of our paper shows the weaknesses of 3DGS and operates under the weaker and more realistic assumption that the first derivative of the images is bounded.
>
> ## R2Q5: Small gains on 1D and 2D tasks.
>
> We agree that the our method's gains are small for toy 1D and 2D super-resolution tasks. However, we demonstrate impressive gains for other 2D task of residual image regression (Tab. 2).
>
>
> ## R2Q6: Analysis of QNN in complex architectures like SR3.
>
> Tab. 1 already shows that adding QNN improves the performance on the super-resolution task with the SR3 architecture. Further detailed analysis— such as exploring how QNN compares to modulation approaches like AdaLN within the SR3 architecture- remains an worthwhile direction, which we reserve for future work.

---

> ### Author Response · Authors · 2025-11-20
> **Response to R2 (cc5r) [2/2]**
>
> ## R2Q7: QNN depends on access to low-frequency signals, limiting generalization to cases without them.
>
> We partly agree that QNN leverages low-frequency signals, but this is a design choice that enhances its versatility, not a fundamental limitation on its application:
>
> - **When available**: QNN directly utilizes the GT low-frequency signal.
> - **When not available**: A pre-computed output or a separate network predicts the low-frequency signal, and the QNN is then applied to model the high-frequency residual. For instance, in NVS experiments, the splatted 3D Gaussian image acts as the low-frequency signal that the QNN refines. This shows QNN's effectiveness even when the GT low-frequency signal is absent.
>
>
> ## R2Q8: Compute overhead.
>
> Let $C_{in}$, $C_{low}$ and $C_{out}$ denote the query dimension, channels of low-frequency input_image and predicted image respectively. Assume $H$ and $W$ are the dimensions of the LF image. Let $L-1$ and $C_h$ denote the number of hidden layers and hidden channels in the neural network.
>
> Layer | Input Tensor | Output Tensor | Multiplications
> -- | - | -|-|
> 0 | $(C_{in}{+}C_{low})\times H \times W$ | $C_{h} \times H \times W$ | $C_{h} \times (C_{in}{+}C_{low}) \times H \times W$
> 1 to L-1 | $C_{h} \times H \times W$ | $C_{h} \times H \times W$ | $C_{h} \times C_{h} \times H \times W$
> L | $C_{h} \times H \times W$ | $C_{out} \times H \times W$ | $C_{h} \times C_{out} \times H \times W$
> Total | - | - | $(C_{in}{+}C_{low}{+}(L-1)C_h{+}C_{out}) \times C_{h} \times H \times W$
>
> Typically, the channels of input_image $C_{low}$ and predicted image $C_{out}$ are both $3$ respectively. Hence, the total multiplications because of QNN is $(C_{in}{+}(L-1)C_h{+}6) \times C_{h} \times H \times W$.
>
> ## R2Q9: Scalability of QNN.
>
> Thanks for raising the fantastic question! We investigated the scalability of QNN in the context of NVS task using the MCMC baseline by varying the number of final Gaussians on the Mip-NeRF 360 dataset. Please see the **Fig. 4 in the updated manuscript**. The results, confirm that adding QNN consistently and reliably improves the NVS PSNR across the entire range of tested Gaussian counts when compared to the MCMC baseline alone.

---

### Official Review · Reviewer_aHPi · 2025-11-02

**Soundness:** 2
**Presentation:** 3
**Contribution:** 2
**Rating:** 4
**Confidence:** 4

**Summary:**

This paper presents "Queried Convolution", a technique for learning high-frequency details from signals with spatial coordinates.

The idea is fairly simple: instead of directly applying a convolutional network to a signal/image, we can first concatenate the signal with encoded queries (eg, coordinates). The authors demonstrate that this is useful for 1D regression, for 2D super-resolution, and as a cleanup/refinement stage for 3D NVS tasks.

*Overall rating:* I'm rating the paper as a weak reject. The execution seems good and the empirical insights will be valuable for the community, but given the weaknesses I'll described below I'm not yet ready to recommend the paper for publication. I'm more than happy to revisit this rating based on response from the authors.

**Strengths:**

The paper is well written, with a conceptually clean and general idea. I found the experiments quite thorough, and convincing for the core message of the paper: CNNs benefit from coordinates concatenated to the inputs.

I appreciated that the authors evaluated QNNs across very different data domains (1D, 2D, and 3D) and provided clear ablations isolating the impact of query inclusion, convolutional context, and encoding type. The NVS application in particular is cool: treating QNN as a geometry-aware refinement stage shows that the method can complement existing generative or reconstruction pipelines.

**Weaknesses:**

As the authors acknowledge in their limitations section, the QNN method is restricted to signals with spatial locality. I don't feel this is a big issue though; the same applies to standard CNNs.

If the error difference is qualitatively perceptible, it would be nice to see analysis on how certain encodings fail, e.g., whether Fourier embeddings alone disrupt spatial smoothness or result in some kind of overfitting to high frequency signals.

I’m slightly worried that the result that vanilla queries outperform Fourier encodings may be misleading, since there are many ways to implement sinusoidal/Fourier features. From my read, the authors try only one variation using random projection matrices with standard deviation 10. If convenient for the authors to run, it would be helpful to see results with:
- Per-axis Fourier features (like in original NeRF)
- Common exponential-style frequency bands (eg, `2**n`) used in original NeRF, Vaswani et al. transformer positional encodings, RoPE, etc.

Concatenating queries with Fourier features appeared to be the best approach for the 1D experiments. Given the ordering of the paper, I found it slightly confusing that the 2D experiments then only evaluated vanilla queries and Fourier feature inputs independently, and that 3D experiments then only use vanilla queries.

If these experiments are difficult to run, it might be helpful to add some discussion on caveats to this result:
> Changing encodings from vanilla to Fourier (Tancik et al., 2020) decreases performance on both val and train sets.


A broader concern I have about the work is that some may find the conclusions of the work, that concatenating signals with coordinates before passing them to convolutional layers is beneficial, not too surprising. I don't think this necessarily detracts from the value of the work (the empirical results are still very useful), but this is in part because this feels like a practice that exists in the field. For example, it's already common to concatenate image inputs with per-pixel ray origins and directions for generative novel view synthesis:
- https://arxiv.org/abs/2405.10314
- https://arxiv.org/abs/2410.17242
- https://arxiv.org/abs/2507.10496
- https://arxiv.org/abs/2501.18804

Are the subset of these works that use CNNs already using QNNs? Discussing connections to past work like these may strengthen the QNN paper.

**Questions:**

Have you considered or compared other approaches for injecting query vectors into the network? For example: AdaLN or FiLM-style approaches might be interesting alternatives to concatenation.

---

> ### Author Response · Authors · 2025-11-20
> **Response to R1 aHPi [1/2]**
>
> We thank the reviewer for the feedback. We first answer the questions in the Questions section followed by the ones listed in the Weaknesses section.
>
> ## R1Q1: Alternatives to concatenating queries into the network such as AdaLN or FiLM-style approaches
>
> Unfortunately, these approaches typically require the network to contain normalization layers (e.g., LayerNorm). Since the current QNN architecture utilized in most tasks **does not include any normalization** layers, we defer the investigation of these alternatives to future work.
>
> ## R1Q2: Do Fourier encodings alone result in some kind of overfitting to high frequency signals?
>
> Method  | | Val| | | Train| |
> -------|-|-|-|-|-|-|
> .|PSNR|SSIM|LPIPS|PSNR|SSIM|LPIPS|
> 3DGS | 27.67 | 0.82 | 0.20 | 29.71 | 0.90 | 0.16 |
> 3DGS + Fourier MLP | 27.78 | 0.82 | 0.20 | 29.77 | 0.90 | 0.16 |
> 3DGS + QNN | **27.96** | **0.83** | 0.20 | **30.11** | 0.90 | 0.16 |
>
> No, Fourier encodings underfit the high frequency signal on the train set. We look at the training PSNR for the NVS task on the Mip-NeRF 360 dataset. As shown in the table above (a breakdown of Tab. 4), **adding a standard Fourier MLP over 3DGS improves the training PSNR marginally**. The negligible gain in training PSNR for the Fourier MLP (29.77) vs. 3DGS (29.71) suggests that Fourier encodings alone lead to underfitting of the high-frequency residual signal. In contrast, QNN achieves a robust increase in training performance (30.11), confirming its superior ability to fit high frequency signals on the training set.
>
>
> ## R1Q3: Comparison with Per-axis Fourier and Exponential Style (NeRF) Encodings
>
> Method | Encoding | | Val| | | Train| |
> -------|-|-|-|-|-|-|-|
> .||PSNR|SSIM|LPIPS|PSNR|SSIM|LPIPS|
> 3DGS | - | 27.67 | 0.82 | 0.20 | 29.71 | 0.90 | 0.16 |
> 3DGS + QNN | Fourier | 27.85 | 0.82 | 0.20 | 29.90 | 0.90 | 0.16 |
> 3DGS + QNN | Per-axis Fourier | 27.85 | 0.82 | 0.20 | 29.89 | 0.90 | 0.16 |
> 3DGS + QNN | Exponential | 27.94 | 0.82 | 0.20 | 29.99 | 0.90 | 0.16 |
> 3DGS + QNN | **Vanilla (Ours)** | **27.96** | **0.83** | 0.20 | **30.11** | 0.90 | 0.16 |
>
>
> Thank you for the excellent suggestion! We compare by integrating these different encodings within the QNN framework and evaluating them for the NVS task on the Mip-NeRF 360 dataset in the above table. The results in the table demonstrates that the Vanilla QNN encoding (Ours) consistently yields the best performance on both the val and train sets. We incorporate these results to extend our Tab. 4.
>
> ## R1Q4: Fig. 1: Concatenating queries with Fourier encodings appeared to be the best approach for the 1D experiments.
>
> There seems to be a confusion about Fig. 1. To clarify, Fig. 1 shows the results with:
>
> Legend | Encoding | Arch | Queries as Input | LF as Input |
> -- | - | - | - | - |
> MLP(query) | Vanilla | MLP | $\checkmark$ | ╳
> MLP(Fourier(query)) | Fourier | MLP |  $\checkmark$ | ╳
> CNN(query) | Vanilla | CNN |  $\checkmark$ | ╳
> CNN (LF) | Vanilla | CNN | ╳ |  $\checkmark$
> QNN(query)| Vanilla | CNN | $\checkmark$ | $\checkmark$
>
> Concatenating **vanilla** encoded queries with the LP signal performs the best for 1D experiments.
>
> ## R1Q5: Do you concatenate Low Pass Signal with Fourier encodings for 2D and 3D experiments?
>
> Yes, we confirm that we **concatentate the Low Signal with the encoded queries (vanilla / Fourier)** for both our 2D and 3D experiments.
> - **2D Residual Image Regression (Tab. 2)**: The inputs to QNN are the concatenation of low-resolution (LR) image and the encoded queries. Rows 4 and 6 detail experiments using the Vanilla and Fourier encodings, respectively.
> - **3D NVS (Tabs. 3 and 4)**: The inputs to the QNN are the concatenation of the 3DGS splatted image (LF signal) and the encoded queries. This is shown in the last row of Tab. 3, and the comparison between encodings is detailed in the 'Encoding' rows of Tab. 4.
>
> ## R1Q6: Validity of changing encodings from vanilla to Fourier decreases performance for NVS
>
> Changing encodings from vanilla to Fourier does indeed lead to decrease in the performance for the NVS task. As detailed in the table below (and in our response to R1Q3), the **Vanilla QNN encoding (Ours) maintains the best** performance across all tested encoding types on the Mip-NeRF 360 dataset.
>
> Change | Encoding | | Val| | | Train| |
> -------|-|-|-|-|-|-|-|
> .||PSNR|SSIM|LPIPS|PSNR|SSIM|LPIPS|
> 3DGS | - | 27.67 | 0.82 | 0.20 | 29.71 | 0.90 | 0.16 |
> 3DGS + QNN | Fourier | 27.85 | 0.82 | 0.20 | 29.90 | 0.90 | 0.16 |
> 3DGS + QNN | Per-axis Fourier | 27.85 | 0.82 | 0.20 | 29.89 | 0.90 | 0.16 |
> 3DGS + QNN | Exponential | 27.94 | 0.82 | 0.20 | 29.99 | 0.90 | 0.16 |
> 3DGS + QNN | **Vanilla (Ours)** | **27.96** | **0.83** | 0.20 | **30.11** | 0.90 | 0.16 |

---

> ### Author Response · Authors · 2025-11-20
> **Response to R1 aHPi [2/2]**
>
> ## R1Q7: Concatenating signals with coordinates exist in Generative NVS literature. This does not detracts from the value of the work, the empirical results are still very useful.
>
> We appreciate the reviewer's insight regarding the existence of signal-to-coordinate concatenation in generative, **sparse-view NVS** literature (e.g., CAT3D, LVSM). We agree that our empirical results stand on their own merit.
>
> We respectfully point out that, to our knowledge, this is the first work to rigorously demonstrate this idea in the context of:
> - **High-Fidelity Refinement**: Using this structure to significantly improve the fidelity of established dense-view NVS methods like 3D Gaussian Splatting (3DGS) and achieve performance levels comparable to methods like Zip-NeRF by focusing on learning high-frequency details (as further discussed in the response to R2Q9).
> - **Theoretical Justification**: Providing a theoretical justification for this specific design choice.
>
> Thus, while the basic operation exists in other literature, its application as a high-frequency learning mechanism for **established NVS benchmarks** is a distinct and valuable contribution.
>
> ## R1Q8: Add four references
>
> We appreciate the suggestion to include additional relevant references. We have updated Section 2 (Related Work) to incorporate PRoPE and MVGD, in addition to CAT3D and LVSM, which were already discussed in the NVS subsection (L142). While PRoPE and MVGD also concatenate images and queries, we note a distinction: they do not employ a convolutional architecture like ours, nor do they specifically demonstrate that this concatenation aids in learning high-frequency signals to boost fidelity.

---

### Author Response · Authors · 2025-11-20
**Rebuttal for Qonvolution**

We thank all the reviewers for their thorough and thoughtful feedback. We are happy that reviewers find QNN has:
- Conceptually clean (R1, R2), simple (R2, R3), general (R1) and unified (R4) idea.
- Thorough experiments (R1, R2) across multiple data domains (R1, R2, R3, R4).
- Elegant theoretical justification (R2, R3, R4).
- Strong gains for 2D regression (R3) and 3D NVS tasks (R2, R3).
- Clear ablations (R1).
- Lightweight and modular (R4), which does not need ray marching or volumetric integration (R4).
- Good literature survey (R3).
- Is well-written (R1).
- Honestly mentioned limitation (R3).

We address the reviewers' insightful concerns in our individual responses. We first answer the questions in the Questions section followed by the ones listed in the Weaknesses section. We mark the changes we made during the rebuttal in blue.

---

### Meta-Review · Area_Chair_YXy8 · 2025-12-22

**Summary:**

This paper introduces Queried-Convolutions (Qonvolutions), which convolves a low-frequency signal with queries to enhance the learning of intricate high-frequency signals. In the first round, this paper received four negative scores (4444). After rebuttal, two reviewers raised their scores from 4 to 6. The reviewers raised concerns regarding the lack of sufficient experimental validation, such as efficiency comparisons and validation on more complex frameworks. While these issues were partially addressed in the rebuttal, they are difficult to be fully resolved within a round of revisions. Therefore, the paper was rejected, but the author is encouraged to revise it and submit it in the next rounds.

**Reviewer Concerns:**

While the authors have partially addressed the reviewers' concerns regarding the experiments, the ability of QNN to handle spectral bias and its computational overhead still require extensive validation.

**Reviewer Scores:**

After full discussion, I believe reviewers will still assign a borderline positive score, as QNN offers advantages in its elegant structure and theoretical analysis. However, the paper's motivation for QNN to reduce spectral bias remains inadequately addressed. This is because while QNN demonstrates overall improvement, the gains in edge regions are not significantly greater than those in smooth regions.

---

### Decision · Program_Chairs · 2026-01-26

Reject